# The PNUTS-PP1 complex acts as an intrinsic barrier to herpesvirus KSHV gene expression and replication

Anne M. Devlin[1], Ashutosh Shukla[1], Julio C. Ruiz[1], Spencer D. Barnes[2], Ashwin Govindan[1], Olga V. Hunter[1], Anna M. Scarborough[1], Iván D'Orso [1] & Nicholas K. Conrad[1]

Control of RNA Polymerase II (pol II) elongation is a critical component of gene expression in mammalian cells. The PNUTS-PP1 complex controls elongation rates, slowing pol II after polyadenylation sites to promote termination. The Kaposi's sarcoma-associated herpesvirus (KSHV) co-opts pol II to express its genes, but little is known about its regulation of pol II elongation. We identified PNUTS as a suppressor of a KSHV reporter gene in a genome-wide CRISPR screen. PNUTS depletion enhances global KSHV gene expression and overall viral replication. Mechanistically, PNUTS requires PP1 interaction, binds viral RNAs downstream of polyadenylation sites, and restricts transcription read-through of viral genes. Surprisingly, PNUTS also represses productive elongation at the 5′ ends of the KSHV reporter and the KSHV T1.4 RNA. From these data, we conclude that PNUTS' activity constitutes an intrinsic barrier to KSHV replication likely by suppressing pol II elongation at promoter-proximal regions.

KSHV is the causative agent of Kaposi's sarcoma, primary effusion lymphoma (PEL), and multicentric Castleman's disease[1]. KSHV switches between distinct transcriptional profiles during its multiphasic life cycle. During latency, it maintains its double-stranded genome in the nucleus of the infected cell and expresses a minimal set of gene products. Upon lytic reactivation, a cascade of gene expression is induced, culminating in the production of infectious virus[2]. KSHV co-opts host factors to transcribe and process its mRNAs and encodes factors that help the virus overcome barriers to gene expression in the host cell. For example, the viral lytic transactivator ORF50 (Rta) induces the switch from latency to lytic phase by driving pol II activity from specific KSHV promoters[3]. Another lytic protein, ORF57 (Mta), promotes expression of KSHV genes by protecting viral RNA from cellular RNA decay pathways[4–9]. Without the support of these viral transcriptional and posttranscriptional factors, many KSHV genes are poorly expressed in mammalian cells.

In addition to regulation at the level of transcription initiation, mammalian cells regulate most genes by controlling pol II elongation[10–13]. A major rate-limiting step in gene expression is the transition from initiating to elongating pol II molecules, resulting in a promoter-proximal pol II pause. To overcome this pause, elongation is promoted by reversible phosphorylation of several factors including the C-terminal domain (CTD) of the largest subunit of pol II, the Negative Elongation Factor (NELF), and the Spt5 subunit of the DRB-sensitivity-inducing-factor (DSIF). Phosphorylation of these targets by P-TEFb promotes release from 5′ pausing by dissociating NELF from the pol II complex and by activating the elongation-promoting activity of DSIF.

Pol II elongation rates also decrease at the 3′ ends of genes in a process closely tied to transcription termination[14,15]. After recognition of the polyadenylation signal (PAS) by the cleavage and polyadenylation specificity factor (CPSF) and cleavage of the nascent RNA, pol II elongation rate slows. In addition, the cleaved nascent RNA becomes subject to exonucleolytic attack by the 5′-to-3′ exonuclease Xrn2. Pol II terminates transcription when Xrn2 reaches the elongating polymerase, which is facilitated by the reduced speed of the polymerase

[1]Department of Microbiology, UT Southwestern Medical Center, Dallas, TX, USA. [2]Lyda Hill Department of Bioinformatics, UT Southwestern Medical Center, Dallas, TX, USA. e-mail: Nicholas.Conrad@UTSouthwestern.edu

within a region called the termination zone. Thus, 3′ end formation and elongation rates are uniquely tied to transcription termination.

The protein phosphatase 1 nuclear targeting subunit PNUTS was recently identified to be required for pol II slowdown in the termination zone[14]. PNUTS binds and regulates the activity of the serine/threonine protein phosphatase 1 (PP1), which was also previously implicated in termination[16–19]. The PNUTS-PP1 complex interacts with pol II and RNA processing factors including the 3′ end formation machinery, and directly binds RNA[20–24]. PNUTS-PP1 occupies genes immediately downstream of the polyadenylation site where it dephosphorylates Spt5 causing accumulation of slowed pol II in the termination zone[14]. This slowdown facilitates termination of the elongation complex[14,15]. Loss of PNUTS-PP1 activity, therefore, results in faster pol II on genes downstream of the transcript end site (TES) and increased pol II occupancy downstream of termination zones[14,25]. Interestingly, PNUTS depletion also leads to a smaller detectable increase in pol II speed within gene bodies, suggesting it may play roles at the promoter-proximal pausing sites, as well as coordinating spliceosome activity with transcription elongation[14,21].

Regulation of KSHV gene expression at the level of transcription elongation remains incompletely understood. Since KSHV uses pol II, it seems likely that similar processes are used by the virus as in host cells. Indeed, at least a subset of KSHV genes are regulated by NELF, DSIF, and P-TEFb[26]. However, differences in the KSHV genes may require distinct modes of elongation regulation. For example, most KSHV genes are intronless, many share poly(A) signals, and they are shorter on average than human genes. Indeed, short genes have differing requirements for specific transcription elongation factors[27]. Moreover, the lack of introns directly affects gene expression since the splicing machinery coordinates the transcription and RNA processing machineries[28–31]. In addition, the KSHV genome is densely packed with genes apparently providing insufficient genomic space to support the multiple kb-long termination zones found in the human genome[32,33]. Given these comparatively unique features of the KSHV genome, it seems probable that regulation of pol II transcriptional elongation and termination are distinct for KSHV genes. However, investigations into these differences have been limited.

Here, we report a CRISPR knockout screen for host factors that negatively regulate the expression of a KSHV reporter gene. Our screen identified PNUTS as a potent suppressor of KSHV gene expression from a heterologously expressed reporter. We further show that depletion of PNUTS robustly increases KSHV transcriptional output upon lytic reactivation in iSLK cells infected with a KSHV bacmid and in patient-derived KSHV-infected PEL cells. Moreover, PNUTS depletion leads to more rapid production of infectious virions from these cells. PNUTS' suppression of the KSHV reporter gene requires its interaction with PP1. Similar to its activities on human genes, PNUTS directly interacts with KSHV RNAs downstream of polyadenylation signals and it promotes termination. However, enhanced expression of our viral reporter gene is driven by a reduction in promoter-proximal pausing after PNUTS depletion. In addition, PNUTS knockdown relieves a promoter-proximal elongation block in the KSHV 1.4 kb transcript (T1.4) in cells latently infected with the KSHV bacmid. From these data, we propose that PNUTS-PP1 suppresses KSHV transcription by promoting 5′ pol II pausing on a subset of KSHV genes and that this activity provides an intrinsic barrier to KSHV gene expression and virus production.

## Results

### A CRISPR screen identifies PNUTS as a suppressor of a KSHV ORF59 reporter gene

To learn more about host factors that suppress KSHV gene expression, we integrated a reporter construct consisting of the KSHV ORF59 gene fused in-frame with GFP into the AAVS1 safe harbor locus in HCT116 cells (Fig. 1a)[34]. ORF59 encodes the viral DNA processivity factor that is highly expressed upon lytic infection but is poorly expressed in heterologous contexts. The KSHV ORF57 protein stabilizes the ORF59 mRNA, so we

screened clonal lines by their ability to upregulate GFP upon expression of ORF57[5,6,8]. We identified several clonal reporter lines in which ORF59-GFP expression is induced by ORF57 and selected one for further use (Supplementary Fig. 1). To perform a genome-wide screen for host factors that limit ORF59 expression, we transduced the selected clonal cell line with the puromycin-selectable lentiviral Brunello sgRNA library[35]. Seven days after transduction, we sorted for the highest GFP-intensity cells and prepared sequencing libraries from both the sorted and unsorted (input) populations (Fig. 1b). Three independent biological replicates of the screen were analyzed by MAGeCK, and the gene encoding the PNUTS protein (PPP1R10) emerged as the top hit (Fig. 1c)[36].

To validate the results of the CRISPR screen, we compared reporter expression after treatment with one of two short-interfering RNAs (siRNAs) targeting PNUTS (siPNUTS 1 and siPNUTS 2) or a non-targeting control (siControl) (Fig. 1d, e). We assayed the steady-state levels of the ORF59-GFP RNA in the clonal reporter line (clone 1), as well as two independent clonal lines with the integrated reporter (Fig. 1f, g). Northern blots confirmed that PNUTS depletion increases the levels of the ORF59-GFP mRNA in all clones (Fig. 1f, g). Thus, the effect of PNUTS depletion is not restricted to the clone used in the CRISPR screen, and the effect with sgRNAs can be reproduced with siRNAs. Given this validation, we pursued a potential role for PNUTS in suppression of KSHV gene expression.

### Depletion of PNUTS broadly enhances KSHV gene expression

We next asked if PNUTS affects ORF59 and other KSHV genes during lytic reactivation. To test this, we used iSLK cells that are latently infected with a bacmid that includes the complete KSHV genome as well as a constitutively expressed GFP as a marker of infection. These cells also carry a doxycycline-inducible copy of ORF50 integrated into the host genome[37,38]. Because the ORF50 transcription factor is necessary and sufficient to induce lytic phase, addition of doxycycline induces lytic reactivation[3]. The histone deacetylase inhibitor sodium butyrate (NaB) further enhances the efficiency of lytic induction. Consistent with our initial results, PNUTS depletion increased ORF59 mRNA levels at 24 h post-induction (hpi) relative to the non-targeting siRNA control (Fig. 2a). We observed increases in additional KSHV transcripts by RT-qPCR at multiple timepoints after lytic induction (8, 24, and 48hpi) (Fig. 2b; Supplementary Fig. 2a).

To gain a more complete picture of the role of PNUTS during the viral lytic gene expression cascade, we performed strand-specific RNA-seq with rRNA-depleted RNA at 0 (uninduced), 8, 24, and 48hpi. We observed robust increases to KSHV gene expression in PNUTS-depleted cells with most genes being upregulated (Fig. 2c). The largest differential expression occurred at 8hpi, suggesting that siPNUTS cells are faster to reactivate but gene expression in control cells may resemble siPNUTS-treated cells over time. We also observed increased viral gene expression in the uninduced samples (0hpi), but it is important to note that the overall expression remained considerably lower in latent phase compared to post-induction in both control and siPNUTS treated cells (Fig. 2d). Genome browser views of representative KSHV genes reflect the same patterns (Supplementary Fig. 2b). Importantly, we observed minimal changes in the expression of ORF50 from either the endogenous or exogenous locus, suggesting that the increased viral gene expression is not simply explained by premature induction of ORF50 expression (Fig. 2c; Supplementary Fig. 2c).

Increased levels of ORF59 have been reported to promote KSHV reactivation[39], so PNUTS knockdown could be driving enhanced gene expression solely by promoting ORF59 expression. To test this hypothesis, we used two independent siRNAs targeting ORF59 to determine if restoring ORF59 RNA levels back to baseline negated the broader effect of PNUTS knockdown. After siRNA treatment, lytic reactivation was induced and RNA was harvested at 24hpi. Knockdown of PNUTS increased the levels of viral lytic genes and co-knockdown of

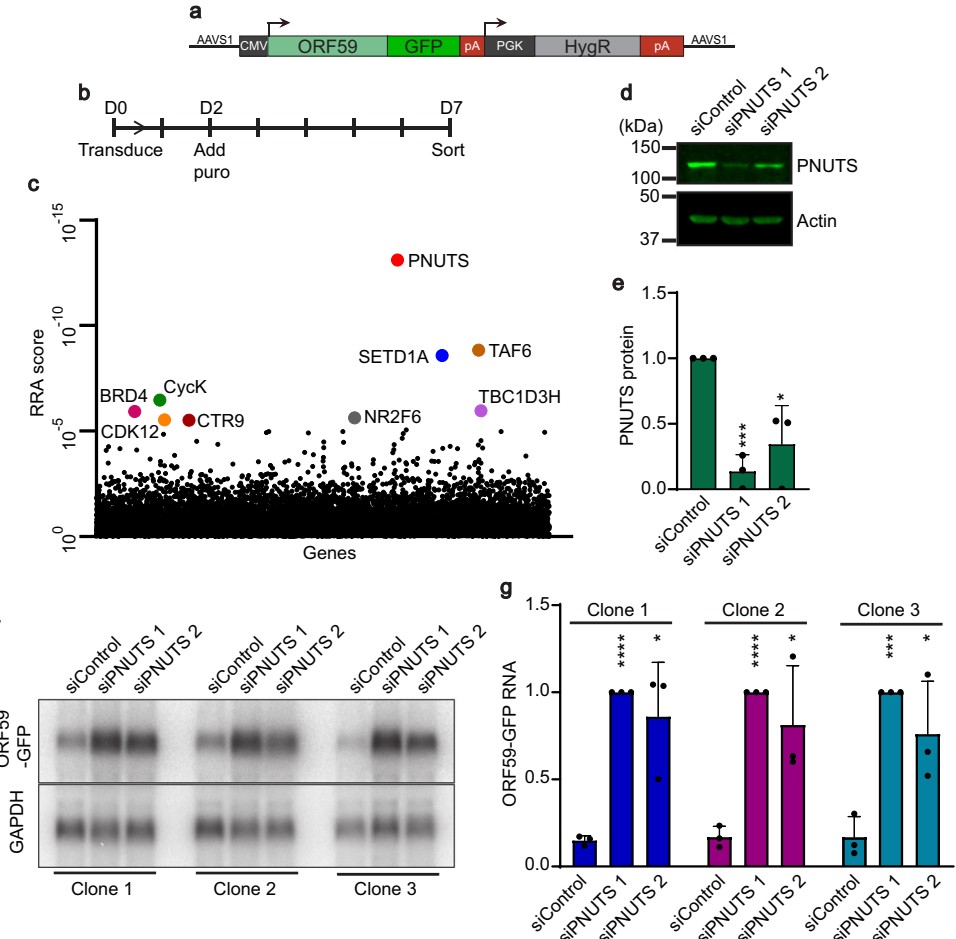

**Fig. 1 | A CRISPR screen identifies PNUTS as a potential suppressor of KSHV ORF59 expression. a** Schematic of ORF59-GFP reporter with flanking AAVS1 homology arms. Dark gray boxes are CMV and PGK promoter regions. PolyA signals, pA (red). **b** Timing of CRISPR knockout screen from lentivirus transduction to sorting, seven days overall with five days of puromycin selection. **c** Gene enrichment plot from three biological replicates of CRISPR knockout screen; genes and non-targeting sgRNAs plotted alphabetically on X-axis. Y-axis is the robust rank aggregation (RRA) scores for each gene. Top nine genes with false discovery rate (FDR) scores <2% denoted with colored dots and labels. **d** Western blot of total protein from the HCT116 reporter line treated siRNAs as indicated. **e** Quantification of PNUTS protein levels after siRNA-mediated knockdown, as in (**d**). Values were first normalized to beta-actin then quantified relative to siControl. Error bars are mean with standard deviation, asterisks denote Student's two-tailed unpaired t-test (siControl vs siPNUTS 1 ***$p = 0.0003$, siControl vs siPNUTS 2 *$p = 0.0185$), ($n = 3$). **f** Northern blot for ORF59-GFP with GAPDH as a loading control using total RNA from three independent clonal reporter lines. **g** Quantification of ORF59-GFP RNA levels after PNUTS depletion in three clones with siPNUTS 1 or siPNUTS 2, as in (**f**). ORF59-GFP was first normalized to GAPDH then quantified relative to siPNUTS 1. Error bars are mean with standard deviation, asterisks denote Student's two-tailed unpaired t-test (Clone 1 siControl vs siPNUTS 1 ****$p = 1\ 7.4 \times 10^{-07}$, siControl vs siPNUTS 2 *$p = 0.0168$; Clone 2 siControl vs siPNUTS 1 ****$p = 1\ 2.0 \times 10^{-05}$, siControl vs siPNUTS 2 *$p = 0.0320$; Clone 3 siControl vs siPNUTS 1 ***$p = 0.0003$, siControl vs siPNUTS 2 *$p = 0.0345$), ($n = 3$). Source data are provided as a Source Data file.

ORF59 did not significantly abrogate this effect (Supplementary Fig. 2d). While it may be the case that increased ORF59 expression helps promote lytic reactivation after PNUTS knockdown, these results indicate that the observed effect of PNUTS on KSHV gene expression is not exclusively due to the increase in ORF59 RNA. Together, these data support the conclusion that PNUTS negatively impacts viral gene expression in the context of KSHV lytic reactivation as well as in reporter genes.

To test the effect of PNUTS depletion in patient-derived KSHV-infected cells, we transduced the primary effusion lymphoma TREx-RTA-BCBL1 cells with lentiviral CRISPR-Cas9 constructs targeting PNUTS or a non-targeting control[40]. Unlike the minimal toxicity we observed from 3-day siRNA-mediated knockdowns of PNUTS in HCT116 or uninduced iSLK cells, high cytotoxicity of sgRNA-mediated PNUTS knockout prohibited use of the TREx-RTA-BCBL1 for knockout experiments (Supplementary Fig. 2e). Therefore, we stably integrated inducible PNUTS (shPNUTS) or non-targeting (shControl) shRNA constructs into TREx-BCBL1 cells[40]. Importantly, these cells lack the

doxycycline-inducible ORF50 transgene, so changes in viral gene expression cannot be an indirect consequence of any effect of PNUTS on the expression of an ORF50 transgene. At bulk levels, we observe only modest decreases in PNUTS protein in the shPNUTS cells (Fig. 2e, f). Nonetheless, diminished PNUTS protein correlated with increases in KSHV gene expression. Northern blotting revealed increased levels of ORF59 mRNA at 0, 24, or 72hpi (Fig. 2g, h). Other lytic genes assayed by RT-qPCR also had similar increases after PNUTS depletion, echoing our results in iSLK cells (Fig. 2i). Notably, after PNUTS depletion in both iSLK and in TREx-shRNA-BCBL1 cells, we observed increases in lytic gene expression prior to reactivation. All latently infected KSHV cell culture lines undergo varying amounts of spontaneous lytic reactivation, so this effect may indicate that there is an increased number of cells that undergo spontaneous reactivation upon PNUTS knockdown, or it may be due to higher levels of gene expression in that subset of cells that undergoes spontaneous reactivation. Alternatively, there could simply be more "leaky" transcription of lytic genes and the cells are not entering full lytic reactivation. In any

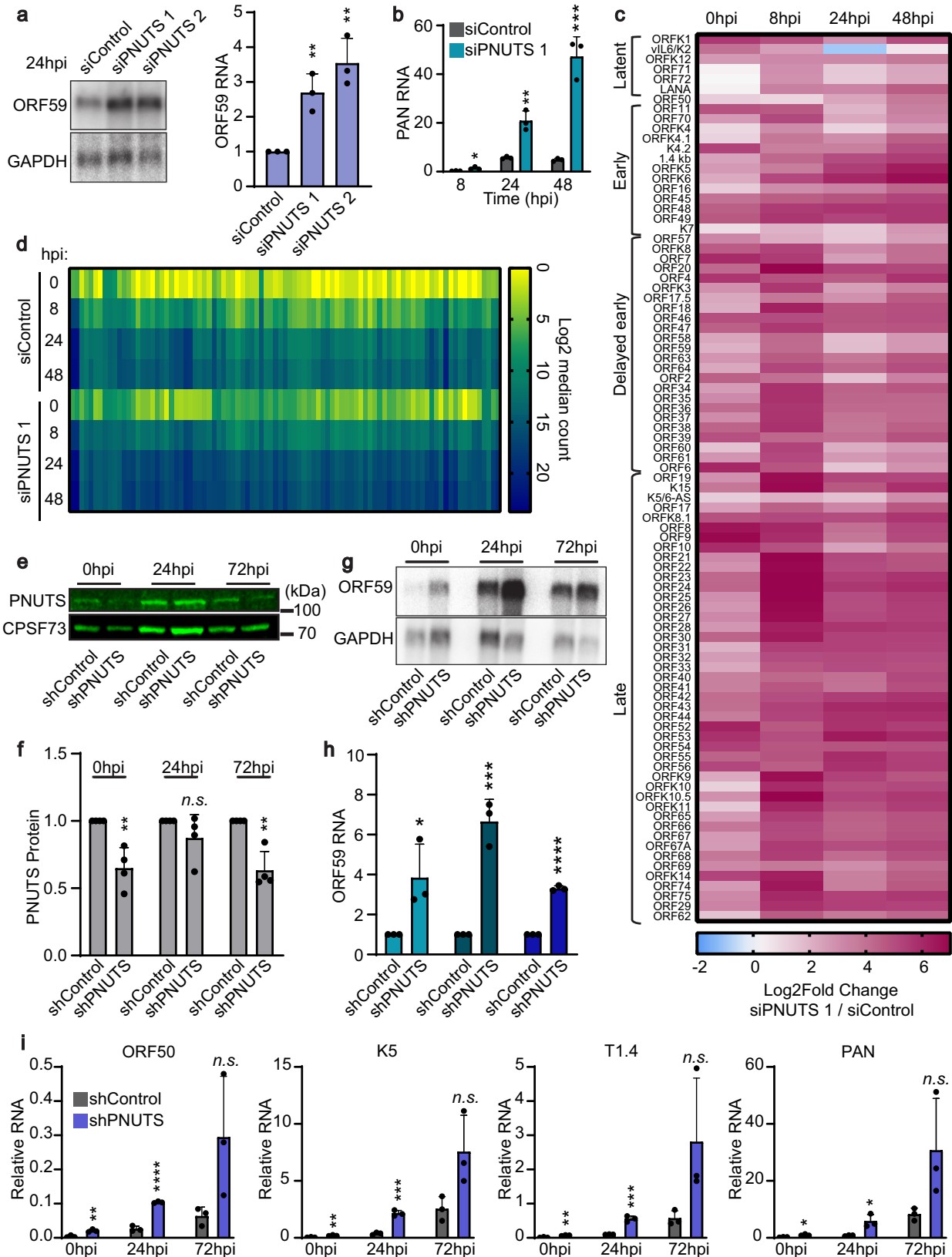

case, our data show that PNUTS restricts viral gene expression in at least two independent KSHV-infected cell lines.

## Depletion of PNUTS increases KSHV viral replication

The dramatic increase in KSHV gene expression after PNUTS knockdown may reflect enhanced viral replication. Alternatively,

these increases may indicate that the normal patterns of regulated viral gene expression are lost, which would then reduce KSHV replication. To determine the effects on overall replication cycle, we first quantified viral DNA (vDNA) levels relative to host DNA in iSLK cells every eight hours after lytic reactivation over a 48-hour time course. Consistent with enhanced replication, we found that

**Fig. 2 | Depletion of PNUTS broadly enhances KSHV gene expression. a** Northern blot of ORF59 with total RNA harvested from iSLK cells at 24 hpi. For quantification, signal was first normalized to GAPDH then quantified relative to siControl. Error bars are mean with standard deviation, asterisks denote Student's two-tailed unpaired t-test: (siControl vs siPNUTS 1 **p = 0.0057, siControl vs siPNUTS 2 **p = 0.0035), (n = 3). **b** RT-qPCR of PAN RNA levels in iSLK cells at 8, 24, and 48 hpi, normalized to beta-actin. Error bars are mean with standard deviation, asterisks denote Student's two-tailed unpaired t-test: (siControl vs siPNUTS 1 8hpi *p = 0.0348, 24hpi **p = 0.0032, 48hpi ***p = 0.0009), (n = 3). **c** Heatmap of Log2-fold changes (Log2FC) in KSHV gene expression in siPNUTS 1 relative to siControl. RNA-seq from three biological replicates in iSLK cells at four timepoints after lytic reactivation. **d** Heatmap of average Log2 median viral gene counts of RNA-seq data from indicated samples. **e** Western blot for PNUTS protein and CPSF73 loading control in TREx-shRNA-BCBL1 lines after 7 days of shRNA induction and 0, 24, or 72 h lytic induction with sodium butyrate (NaB). **f** Quantification of relative PNUTS protein levels at multiple timepoints of lytic induction for shControl and shPNUTS cell lines; values were first normalized to CPSF73 and quantified relative to

shControl. Error bars are mean with standard deviation, asterisks denote Student's two-tailed unpaired t-test, n.s. not significant: (shControl vs shPNUTS 0hpi **p = 0.0036, 24hpi p = 0.1952, 72hpi **p = 0.0019), (n = 4). **g** Northern blot of ORF59 expression from TREx-shRNA-BCBL1 shControl and shPNUTS cells at 0, 24, and 72hpi. **h** Quantification of ORF59 northern blot data from TREx-shRNA-BCBL1 cells as in (**g**). Error bars are mean with standard deviation, asterisks denote Student's two-tailed unpaired t-test: (shControl vs shPNUTS 0hpi *p = 0.0128, 24hpi ***p = 0.0009, 72hpi ****p = 6.3 × 10$^{-6}$), (n = 3). **i** RT-qPCR of lytic genes at 0, 24, and 72hpi from TREx-shRNA-BCBL1 lines, normalized to beta-actin mRNA. Note that expression of several KSHV lytic genes peaks at ~24–48hpi, while relative expression is highest at 72hpi, likely due to host shut-off. Error bars are mean with standard deviation, asterisks denote Student's two-tailed unpaired t-test: (shControl vs shPNUTS ORF50 0hpi **p = 0.0049, 24hpi ****p = 9.3 × 10$^{-5}$, 72hpi n.s. p = 0.0905; K5 0hpi **p = 0.0078, 24hpi ***p = 0.0004, 72hpi n.s. p = 0.0614; T1.4 0hpi **p = 0.0048, 24hpi ***p = 0.0007, 72hpi n.s. p = 0.1073; PAN 0hpi *p = 0.0118, 24hpi *p = 0.0202, 72hpi n.s. p = 0.1011), (n = 3). Source data are provided as a Source Data file.

depletion of PNUTS increased vDNA content early in lytic phase (Fig. 3a).

To monitor infectious virus production, we induced reactivation in siPNUTS or siControl treated iSLK cells and harvested supernatants at 48hpi. Supernatants were filtered, benzonase treated to degrade free DNA, and used to infect HCT116 cells. Virion production was gauged by flow cytometry for GFP expressed from the KSHV bacmid (Fig. 3b). We observed a striking increase (>10-fold) in virus production from the PNUTS-depleted cells at 48hpi (Fig. 3c). Moreover, we detected infectious virus produced in PNUTS-depleted cells at 24hpi (Fig. 3d, e; Supplementary Fig. 3). In our hands, virus production from induced iSLK cells typically appears ~72–96hpi, so observation of virus production as early as 24hpi was quite unexpected. To test virus production in the TREx-shRNA-BCBL1 cells, we performed a similar experiment except that we assayed transcript levels of the latent gene LANA in the recipient cells by RT-qPCR (Fig. 3f). Once again, we observed an increase in infectious virus upon PNUTS depletion in these patient-derived infected cells (Fig. 3g). These results reveal a previously unknown relationship between PNUTS and KSHV and suggest that PNUTS acts as an intrinsic barrier to KSHV gene expression and replication.

### PNUTS' suppression of viral gene expression correlates with its PP1-binding and transcription activities
We next sought mechanistic insights into PNUTS' suppression of KSHV gene expression. Because ORF59 is one of several KSHV RNAs targeted by nuclear RNA quality control pathways, we tested whether PNUTS may be functioning in these known pathways. To do so, we co-depleted PNUTS with RNA decay factors known to target ORF59 mRNA (Fig. 4a)[5]. Co-depletion of PNUTS with ZFC3H1 or MTR4 significantly increased ORF59-GFP RNA levels above those seen in each of the individual knockdowns. Interestingly, co-depletion of PNUTS and ARS2 did not have an additive effect (Fig. 4a). This is notable because ARS2 is not only a decay factor, but like PNUTS, it has been linked to transcription termination[41–43]. ARS2 depletion in iSLK cells enhances KSHV gene expression but, unlike PNUTS depletion, it does not enhance viral replication[44]. These data imply a role for PNUTS in viral gene expression distinct from RNA-decay pathways that post-transcriptionally target KSHV RNAs.

In mammalian transcription, PNUTS functions through its interaction with PP1, so we tested if this interaction is required for suppression of KSHV gene expression. To do so, we transiently co-transfected HEK293A-TOA cells with an ORF59-GFP reporter and empty vector control, an siRNA-refractory N-terminally Flag-tagged wildtype PNUTS (FL-PNUTS), or Flag-tagged PNUTS with the same siRNA-refractory mutations and a point mutation that abrogates PNUTS-PP1 binding (FL-W401A)[23] (Supplementary Fig. 4a). As expected, knockdown of PNUTS increased ORF59-GFP mRNA expression,

and this effect was rescued by the addition of WT PNUTS. However, the PP1-binding mutant PNUTS was not able to rescue the phenotype (Fig. 4b). Thus, the PNUTS mechanism for suppression of the ORF59 reporter requires its established regulatory role with PP1.

PNUTS-PP1 functions during pol II transcription in eukaryotic cells, so we wondered if it acted with other pol II regulators identified by our CRISPR screen[14,20,21,23,25,45]. Along with PNUTS, our screen identified the histone methyltransferase SETD1A, the cyclin-dependent kinase CDK12, and its associated cyclin partner cyclin K (CycK) (Fig. 1c). Interestingly, SETD1A is recruited to transcription start sites by interaction with the major PNUTS-interaction partner WDR82 and is also known to interact with CycK[46–48]. Like PNUTS depletion, loss of WDR82, PP1, or the SET1 H3K4 methyltransferase induces termination defects[25]. The CDK12-CycK complex is involved in regulation of pol II elongation and termination as well[27,49]. Depletion of SETD1A, CDK12, or Cyclin K with siRNAs all increased levels of ORF59-GFP in our reporter cells, validating our CRISPR results (Fig. 4c and Supplementary Fig. 4b). However, co-depletion of PNUTS with these factors did not increase ORF59-GFP levels beyond PNUTS depletion alone, indicating that PNUTS may function in the same pathway as these pol II regulators (Fig. 4c). We also note that loss of CDK12 has been shown to induce the premature cleavage and polyadenylation of specific transcripts, including PNUTS[27]. As such, the observed phenotypes of CDK12/Cyclin K depletion may be driven by the loss of PNUTS, or they may indicate a potential regulatory relationship between these factors. Together these data indicate that PNUTS functions with PP1 to downregulate ORF59 levels and suggest potential mechanistic overlap in this activity between PNUTS-PP1 and other transcriptional regulators identified in our screen.

### PNUTS suppresses transcription readthrough in KSHV
To investigate a potential role of PNUTS in pol II transcription of KSHV genes, we tested whether loss of PNUTS induces similar transcriptional aberrations in KSHV genes as observed in mammalian genes. PNUTS-PP1 decreases pol II transcription rates after a cleavage and polyadenylation site has been recognized, which in turn increases pol II termination efficiency[14]. Therefore, we tested whether PNUTS-PP1 depletion leads to increased levels of transcription downstream of viral genes. We used our ORF59-GFP reporter line for these assays because direct observation of read-through transcription on the KSHV genome is complicated by the dense clustering of transcription units. Using northern blots of poly(A)-selected RNA probed for ORF59 or the downstream hygromycin resistance gene (Fig. 5a), we observed that PNUTS depletion leads to the accumulation of an extended read-through transcript. The length of the transcript corresponds to an elongated transcript spanning from the ORF59-GFP promoter through the hygromycin poly(A)signal (Fig. 5b). The hygromycin resistance mRNA is concomitantly downregulated in

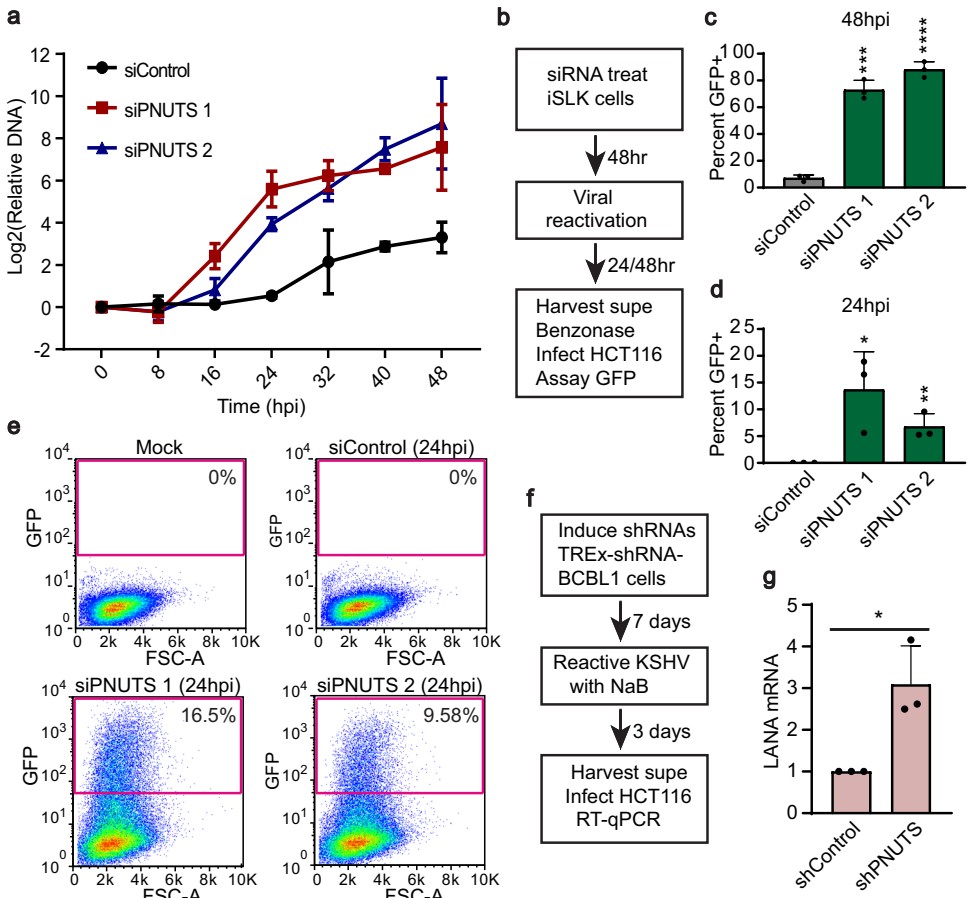

**Fig. 3 | Depletion of PNUTS enhances KSHV viral replication. a** vDNA content relative to host genome was quantified by qPCR at various timepoints from 0 to 48hpi in siControl, siPNUTS 1 or siPNUTS 2-treated cells. Error bars are mean with standard deviation, ($n = 3$). **b** Schematic of viral replication assay from iSLK cells after PNUTS depletion. **c** Quantification of the percent of recipient cells infected with virus produced from non-targeting control versus PNUTS-depleted cells after 48 h of lytic induction. Error bars are mean with standard deviation, asterisks denote Student's two-tailed unpaired t-test: (siControl vs siPNUTS 1 ***$p = 0.0001$, siControl vs siPNUTS 2 ****$p = 2.4 \times 10^{-5}$), ($n = 3$). **d** as in **c**, except from virus harvested at 24hpi from iSLK cells. Error bars are mean with standard deviation, asterisks denote Student's two-tailed unpaired t-test: (siControl vs siPNUTS 1

*$p = 0.0289$, siControl vs siPNUTS 2 **$p = 0.0087$), ($n = 3$). **e** Representative flow cytometry results of recipient HCT116 cells after infection assay. **f** Schematic of TREx-shRNA-BCBL1 virus production assay. Expression of the shRNAs was first induced by treatment with doxycycline, and lytic reactivation was induced by sodium butyrate. **g** LANA expression as determined by RT-qPCR in recipient cells 24 h after infection with supernatant from shControl or shPNUTS TREx-shRNA-BCBL1 cells. Signal was first normalized to beta-actin and is displayed relative to shControl. Error bars are mean with standard deviation, asterisk denotes Student's two-tailed unpaired t-test: *$p = 0.0172$, ($n = 3$). Source data are provided as a Source Data file.

the siPNUTS cells, presumably due to promoter occlusion by read-through transcription[50]. These data support a similar role for PNUTS in pol II termination in our reporter as that observed in host genes.

PNUTS is an RNA binding protein, so we next investigated whether PNUTS binds viral RNAs during lytic reactivation[19,21,24]. We performed eCLIP using antibodies recognizing endogenous PNUTS to immuno-precipitate crosslinked RNA-protein complex from iSLK cells at 24hpi. PNUTS binding was enriched on viral RNAs, but this does not necessarily reflect a specificity for viral transcripts since most new transcription at this time point is viral due to KSHV host shutoff mechanisms[51]. The eCLIP assay uses a size-matched input (SMI) as a reference point, representing total input sample that is isolated in parallel with the immunoprecipitated samples after size separation on SDS-PAGE[52]. Comparison of immunoprecipitated (IP) signal to the SMI revealed crosslinking of PNUTS to regions immediately 3′ of the viral transcript end sites (TES). For example, IP and SMI coverage across the ORF40/41 locus is similar within the gene body but enhanced after the TES (Fig. 5c, compare purple and gray). Moreover, metagene plots confirm preferential PNUTS binding immediately downstream of viral TESs (Fig. 5d). We also noted an apparent increase in RNA-Seq reads

mapping to the termination zones after PNUTS depletion (Fig. 5c, compare blue and black). This is unlikely to exclusively reflect the higher transcript levels since metagene analysis supports an increase in RNA abundance immediately downstream of viral TESs in the siP-NUTS samples (Fig. 5e; Supplementary Fig. 5). Together, these observations suggest that PNUTS serves similar roles in transcription termination of viral genes as it does with host genes.

## PNUTS promotes 5′ pol II pausing on the ORF59-GFP viral reporter

While our data support a role for PNUTS in viral transcription termination, it is not immediately obvious how this activity would suppress viral gene expression. In one hypothesis, PNUTS-induced slowdown of pol II downstream of viral TESs leads to a "traffic jam" that slows pol II along the viral gene. Alternatively, PNUTS may impede transcription elongation along viral genes by affecting promoter clearance. To distinguish these hypotheses, we monitored RNA synthesis by RNA labeling and chromatin immunoprecipitation (ChIP) assays. We could not use the reactivated virus or our GFP reporter because the highly transcribed genes are too closely spaced to resolve pol II occupancy from upstream

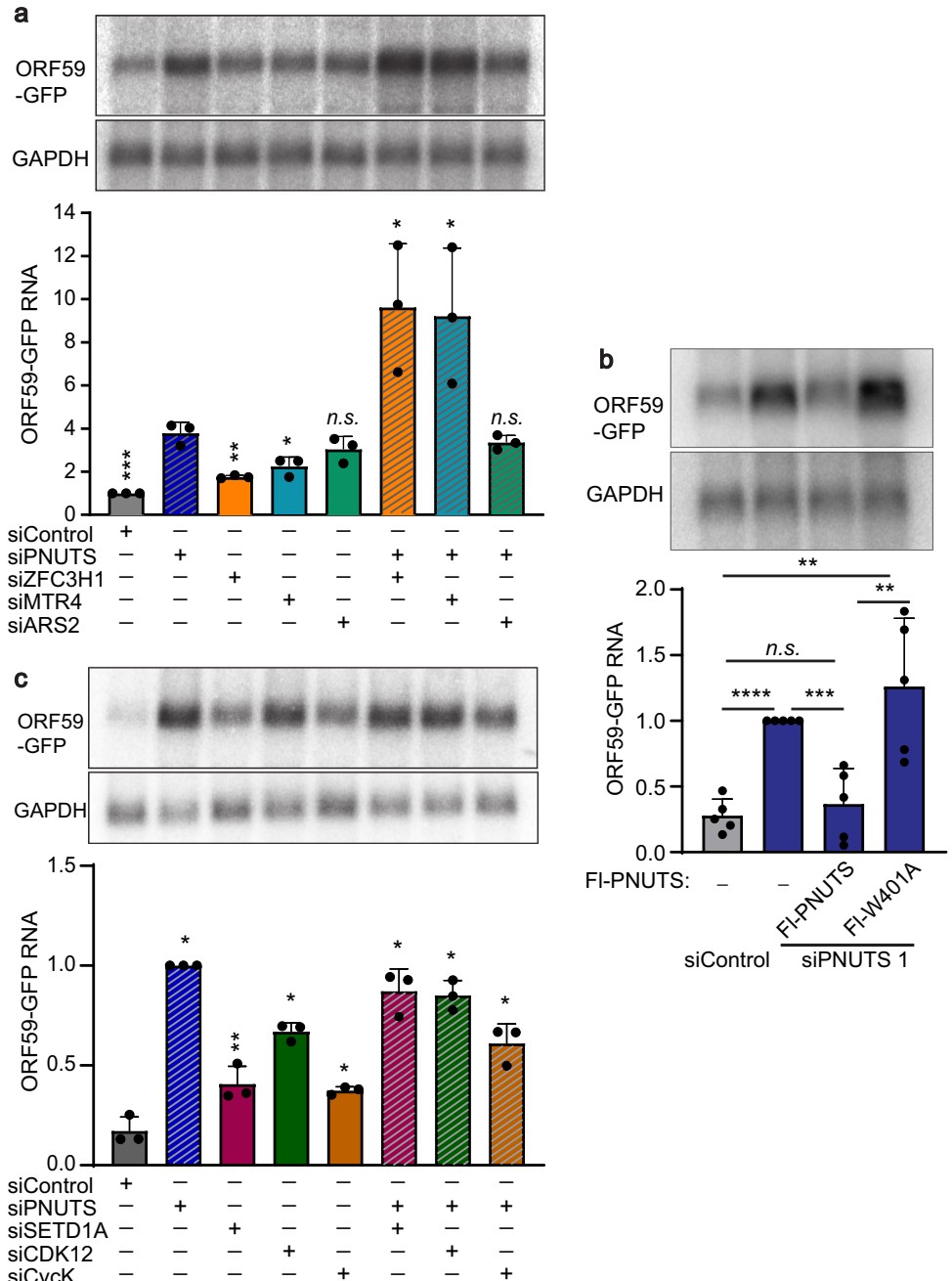

**Fig. 4 | PNUTS-mediated suppression of ORF59-GFP correlates with its PP1-dependent transcriptional functions. a** Representative northern blot and quantification of ORF59-GFP from the reporter line treated with the indicated siRNAs. Values were normalized to GAPDH and quantified relative to siControl. Error bars are mean with standard deviation, asterisks denote Student's two-tailed unpaired t-tests compared to single siPNUTS knockdown: (siControl ***$p$ = 0.0007, siZFC3H1 **$p$ = 0.0025, siMTR4 *$p$ = 0.0164, siARS2 *n.s.* $p$ = 0.1733, siPNUTS/siZFC3H1 *$p$ = 0.0278, siPNUTS/siMTR4 *$p$ = 0.0428, siPNUTS/siARS2 *n.s.* $p$ = 0.2810), ($n$ = 3). **b** Representative northern blot and quantification of ORF59-GFP RNA from HEK293A-TOA cells treated with siRNAs and then transiently transfected with pAAVS1-ORF59-GFP-Hyg and empty vector, Fl-PNUTS, or Fl-W401A (both with siRNA-refractory silent mutations). RNA was harvested at 72 h post-transfection

with siRNAs and 24 h post-transfection of DNA plasmids. ORF59-GFP RNA was normalized to GAPDH and is graphed relative to siPNUTS with empty vector. Error bars are mean with standard deviation, asterisks denote Student's two-tailed unpaired t-test: (siControl vs siPNUTS 1 ****$p$ = 1.4 × 10⁻⁶, siControl vs Fl-PNUTS *n.s.* $p$ = 0.5284, siControl vs Fl-W401A **$p$ = 0.0092, siPNUTS 1 vs FL-PNUTS ***$p$ = 0.0008, Fl-PNUTS vs Fl-W401A **$p$ = 0.0092), ($n$ = 5). **c** Representative northern blot and quantification of ORF59-GFP RNA, as in (**a**). Values were normalized to GAPDH and relative to siPNUTS. Error bars are mean with standard deviation, asterisks denote Student's two-tailed unpaired t-test compared to single siControl sample: (siPNUTS *$p$ = 0.0116, siSETD1A **$p$ = 0.0032, siCDK12 *$p$ = 0.0132, siCycK *$p$ = 0.0370, siPNUTS/siSETD1A *$p$ = 0.0116, siPNUTS/siCDK12 *$p$ = 0.0158, siPNUTS/siCycK *$p$ = 0.0160), ($n$ = 3). Source data are provided as a Source Data file.

versus downstream genes. Therefore, we constructed a new reporter that eliminates the hygromycin promoter (PGK) and inserts a segment of non-expressed KSHV genomic sequence (Fig. 6a; K11). After integration into the AAVS1 site of HCT116 and clonal line selection, we verified

that PNUTS depletion yields robust increases of ORF59-GFP RNA (Supplementary Fig. 6a). To test pol II activity, we performed 4SU-labeled RNA capture to compare the levels of RNA synthesized during a 10-minute 4SU pulse. We observed enhanced levels of newly made RNA at

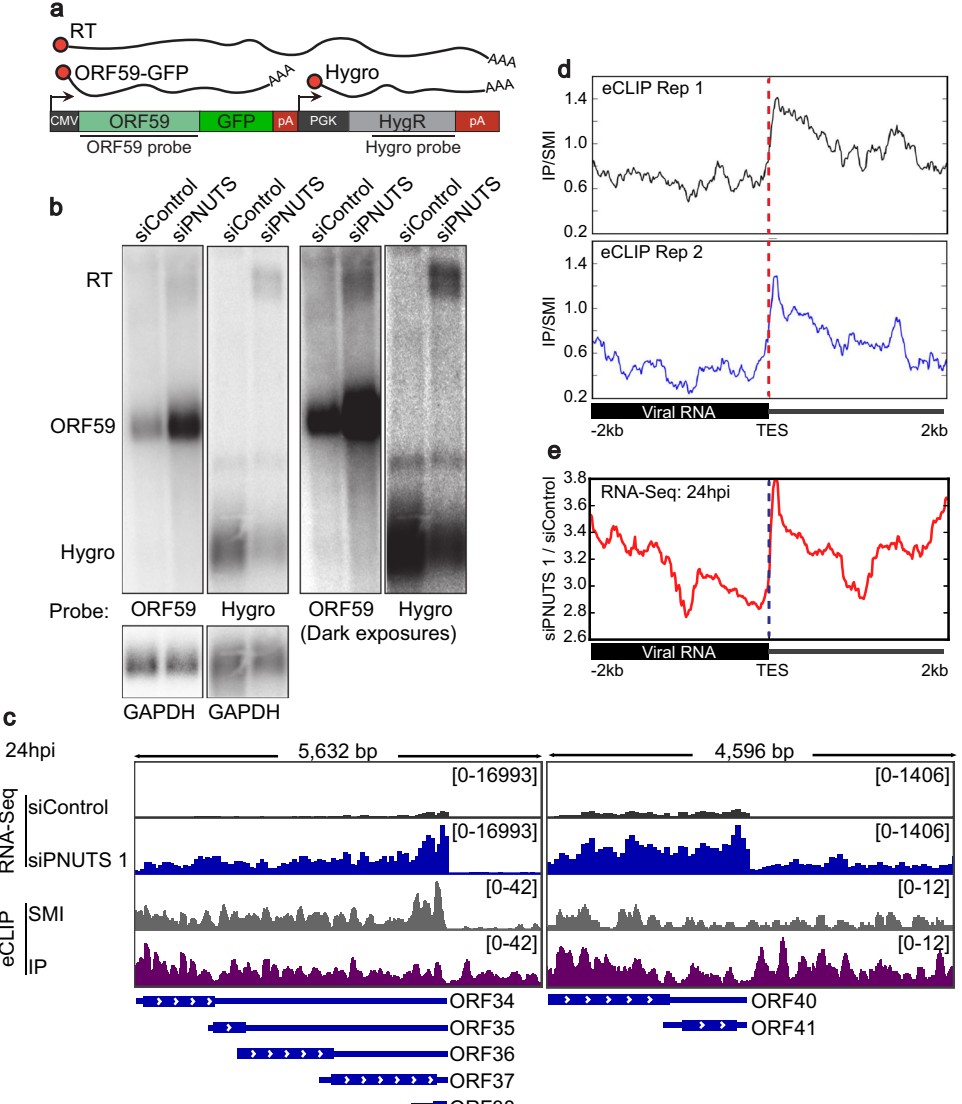

**Fig. 5 | PNUTS binds to KSHV RNA and suppresses readthrough transcription in KSHV. a** Schematic of integrated reporter construct with northern probes and predicted RNA species listed. **b** Northern blots of polyA-selected mRNA from HCT116 reporter cells, using a cocktail of both PNUTS siRNAs. The blots on the right are overexposed versions of the same blots on the left. **c** Representative IGV browser images of KSHV gene coverage from RNA-Seq and PNUTS eCLIP in iSLK cells at 24hpi. Dark gray is siControl, dark blue is siPNUTS. Light gray is the size-matched input (SMI) sample for eCLIP, and purple is the eCLIP of endogenous PNUTS in iSLK cells at 24hpi. **d** Metagene plots of two independent eCLIP experiments for PNUTS in iSLK cells, 24hpi. The graphs show the levels of PNUTS-bound viral RNA relative to SMI, centered on TESs. **e** Metagene as in (**c**), with RNA-seq data from one biological replicate at 24hpi. Source data are provided as a Source Data file.

two amplicons along the reporter with two different anti-PNUTS siRNAs, although the second siRNA has a less dramatic effect, consistent with its reduced knockdown efficiency (Fig. 6a, b; Fig. 1d). These results support the idea that enhanced pol II activity on viral genes leads to the increase in steady-state RNA levels, but they do not distinguish between the traffic jam and promoter clearance models.

To make this distinction, we performed two independent biological replicates of ChIP experiments using an antibody targeting the pol II RBP3 subunit. We used *Drosophila* spike-in chromatin to normalize ChIP-seq with both PNUTS siRNAs and control. Consistent with published studies demonstrating that PNUTS causes pol II slow-down after TESs, PNUTS depletion led to reduction of pol II peaks downstream of human gene TESs and increased pol II occupancy further downstream (Fig. 6c, d). PNUTS-depleted cells had minimal reductions in pol II promoter and gene body occupancy (Fig. 6e; Supplementary Fig. 6b). However, the pol II profiles on the reporter are distinct from those on most human genes. First, we do not detect a clear pol II pause site downstream of either of the poly(A) sites (Fig. 6f; Supplementary Fig. 6c). Second, and more importantly, the promoter-proximal pol II peak is reduced, and pol II density increases in the gene body upon PNUTS depletion. Indeed, the pause index (PI), which quantifies reads in the promoter-proximal (PP) region relative to those in the promoter distal (PD) region, is considerably lower in our reporter (Fig. 6g; Supplementary Fig. 6d). These data strongly support the conclusion that PNUTS restricts ORF59-GFP gene expression by contributing to promoter-proximal pausing. As a result, loss of PNUTS activity increases pol II activity along the gene thereby enhancing gene expression.

Consistent with previous reports, the lowered PI was not evident for host genes at the global level (Fig. 6h)[14,25,53]. In fact, a small but statistically significant increase in PI was observed only for siPNUTS 2, but the biological relevance of this is questionable. Since viruses often usurp mechanisms from their hosts, we examined metagene profiles

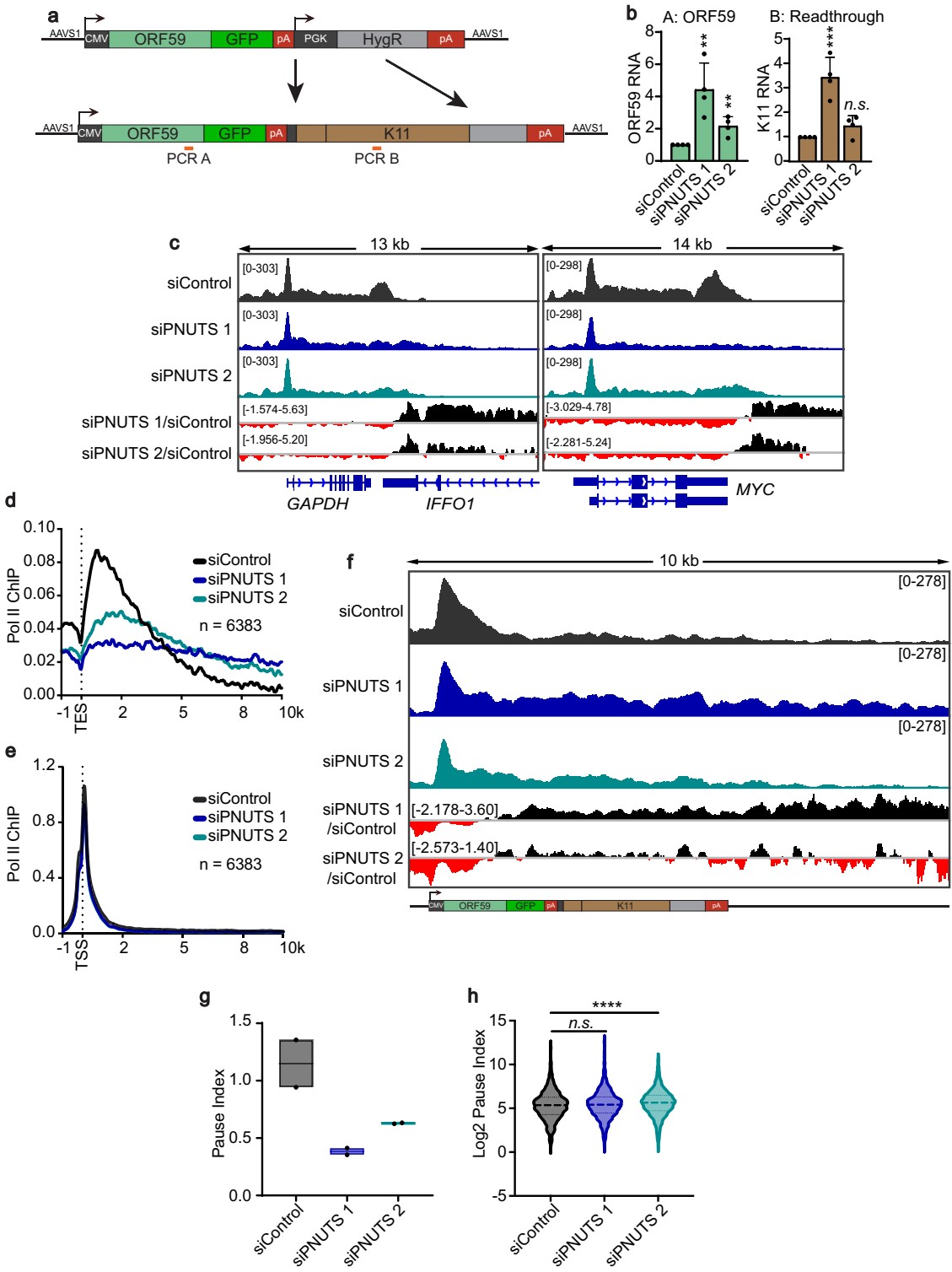

for host genes sharing features of many viral genes. Analysis of intronless genes, short protein-coding genes (<6 kb), and lincRNAs revealed no clear similarities in pol II occupancy to the reporter (Supplementary Fig. 6e–g). Thus, further work is required to determine whether specific host genes require PNUTS for promoter-proximal pausing.

### PNUTS promotes 5´ pol II pausing on the T1.4 KSHV gene

The unexpected 5´ pause effect we observed on our integrated reporter prompted us to ask if we could observe this activity of PNUTS

directly on the viral genome. As previously noted, the density of transcription units and high levels of pol II occupancy during lytic replication prevent analysis of pol II profiles on the viral genome by ChIP-Seq. However, prior studies have identified the presence of promoter-proximally paused pol II on some KSHV genes during latency, when overall pol II density on the viral genome is very low[26]. Additionally, we have observed that depletion of PNUTS increases the expression levels of some KSHV RNAs before lytic induction (Fig. 2c, d). Therefore, we performed pol II ChIP-Seq in latently infected iSLK cells after PNUTS knockdown. These experiments revealed a 5´ pause

**Fig. 6 | PNUTS depletion increases transcriptional output of a viral gene reporter by decreasing promoter-proximal pausing. a** Schematic of the modified reporter used for ChIP-seq and 4SU experiments. *A* and *B* mark the 4SU-qPCR amplicon locations. **b** RT-qPCR of 4SU-labeled RNA following a 10 min 4SU pulse before sample harvesting and 4SU selection. Amplicon levels were normalized to 4SU-labeled beta-actin and quantified relative to siControl. Error bars are mean with standard deviation, asterisks denote Student's two-tailed unpaired t-test: (ORF59 siControl vs siPNUTS 1 **$p = 0.0060$, siControl vs siPNUTS 2 **$p = 0.0066$; Readthrough siControl vs siPNUTS 1 ***$p = 0.0009$, siControl vs siPNUTS 2 *n.s.* $p = 0.0684$), ($n = 3$). **c** Representative IGV images of human genes GAPDH and MYC ChIP-sequencing profiles in non-targeting control (dark gray), siPNUTS 1 (dark blue), and siPNUTS 2 (teal). Bottom tracks are relative (Log2 ratio) pol II occupancy comparing PNUTS-depleted samples to non-targeting control; black and red denote higher and lower signal in PNUTS-depleted samples, respectively. **d** Metagene centered on TES for uniquely annotated ($n = 6383$) protein-coding

genes at least 1 kb long. Due to extensive signal from pol II readthrough of adjacent genes, we filtered our dataset of protein coding genes to include only those that have ≥10 kb flanking the neighboring genes. **e** Same as **d**, except metagenes centered on the transcription start site (TSS). **f** IGV browser image of ChIP-seq reads from the integrated reporter construct. Labeling as in panel (**c**). **g** Box plot showing the PI of the integrated reporter under the indicated siRNA treatments. Black line inside the boxplot indicates the mean values. The black dots show data points from two replicates. **h** Violin plot showing the PI (Log2 scale) of human protein-coding genes ($n = 1358$) under the indicated siRNA conditions. Thick dashed line indicates median, thin dashed lines below and above show 1st and 3rd quartile respectively. Genes (as in **d**) having low read coverage in the promoter-proximal region were excluded from the analysis. Two-sided Wilcoxon rank sum tests were used for pairwise comparisons, (siControl vs siPNUTS 1 *n.s.* $p = 0.0975$, siControl vs siPNUTS 2 ****$p < 0.0001$). Source data are provided as a Source Data file.

release pattern mirroring that observed on our integrated reporter for the KSHV T1.4 RNA (Fig. 7a). While its function remains unclear, T1.4 is transcribed from the origin of lytic replication locus (oriLyt) and is essential for replication[54,55]. The T1.4 ORF contains two direct repeats where ChIP-Seq reads cannot be mapped (Fig. 7a, DR1 and DR2), so to calculate pol II occupancies in the promoter compared to the gene body we selected a region outside of the direct repeats but upstream of both alternative polyadenylation sites for the T1.4 RNA (Fig. 7a top, PP and PD, red). In siPNUTS-treated cells, we observed increased pol II occupancy in both PP and PD regions, but a more marked increase in the latter (Fig. 7a, b). Indeed, we find that both PNUTS siRNAs decrease the pause index for T1.4, indicating that PNUTS affects 5′ elongation blocks on at least one other virally encoded gene, and this mechanism is operative in the context of the latent viral genome (Fig. 7c).

RNA-Seq data and RT-qPCR data from uninduced iSLK cells and TREx-shRNA-BCBL1 cells confirm increased levels of the T1.4 RNA after PNUTS depletion (Fig. 7d, e; Supplementary Fig. 7a). Importantly, PNUTS does not appear to affect promoter pausing on all viral genes. ChIP-Seq profiles of the latency locus in uninduced cells do not display the same altered 5′ pause profile, and do not show increased expression after PNUTS knockdown (Supplementary Fig. 7b, c; Fig. 2c). Instead, these genes reflect the majority of host genes, with minimal changes within the gene body but increased readthrough transcription downstream of the polyA site (Fig. 6d, e; Supplementary Fig. 7b, d). Our data strongly suggest that PNUTS functions in viral gene termination similar to host genes, but additionally plays a suppressive role on specific KSHV genes by promoting promoter-proximal pausing (Fig. 7f).

## Discussion

The complex mechanisms of mammalian transcription, RNA processing, and translation create intrinsic barriers for viral gene expression within the host cell. Although these barriers did not necessarily evolve to thwart viral propagation, viruses must regulate their own gene expression within this cellular context. For example, the nonsense-mediated RNA decay (NMD) pathway likely evolved to protect cells from toxic effects of truncated proteins, but NMD presents a barrier to both DNA and RNA viruses[56,57]. Based on the work presented here, we propose that the PNUTS-PP1 complex is a previously unidentified intrinsic barrier to KSHV gene expression. Mechanistically, we show that PNUTS-PP1 is important for proper termination of KSHV genes, similar to its defined roles in human gene expression (Fig. 7f, top). In addition, we found that the up-regulation of T1.4 and ORF59 genes results from PNUTS-PP1 restriction of pol II at the 5′ ends of the genes (Fig. 7f, bottom). The extent of this promoter-proximal region pausing in other viral and host genes remains unknown, but we think it is likely that other viral and host genes are targeted in this fashion. Given its suppressive role, it is possible that KSHV evolved mechanisms to actively repress PNUTS activity. In fact, PNUTS was found to bind KSHV

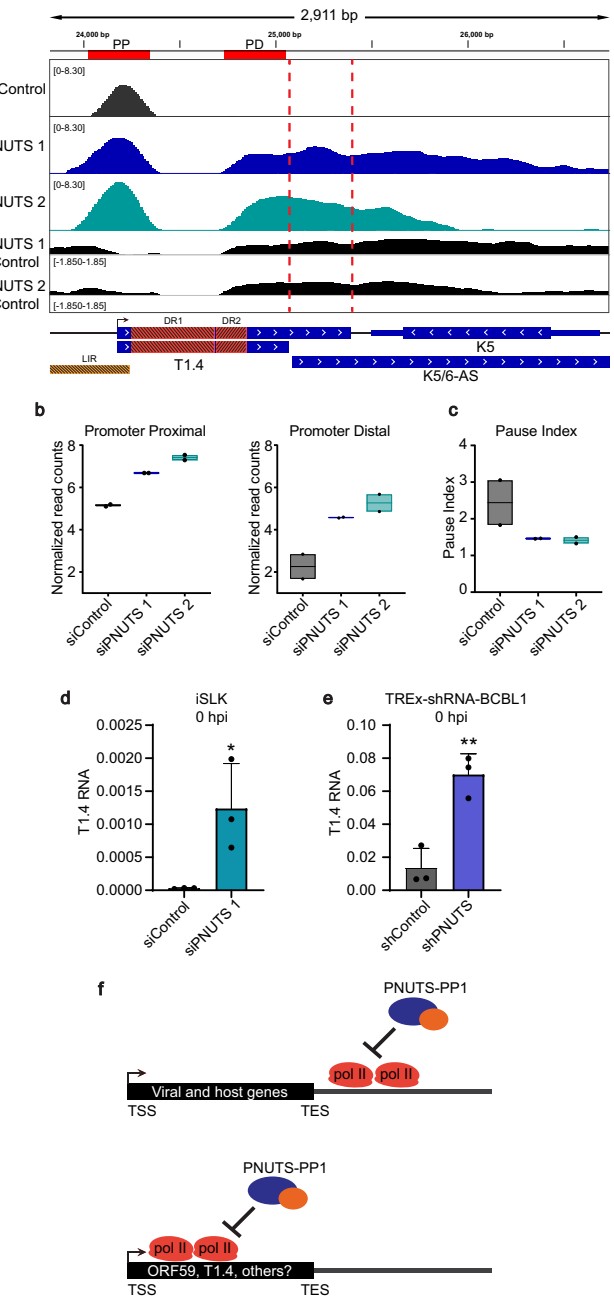

**Fig. 7 | PNUTS depletion increases expression of the viral T1.4 RNA by decreasing promoter-proximal pausing. a** IGV browser view of pol II ChIP-Seq in uninduced iSLK cells at the T1.4 locus. Track profiles as in Fig. 6c. LIR denotes a long inverted repeat region, DR1 and DR2 denote two direct repeat regions. The two polyadenylation sites for the long and short isoforms of T1.4 are denoted with red dotted lines. The regions defined as PP and PD used for PI determination are shown in red at the top of the diagram. **b** Boxplots of input normalized pol II ChIP read counts; PP and PD regions defined in (**a**). Black line inside the boxplot indicates the mean values. The black dots show data points from 2 replicates. **c** Boxplot of PI in control or PNUTS knockdown cells for T1.4. Black line inside the boxplot indicates the mean values. The black dots show data points from 2 replicates. **d** RT-qPCR of T1.4 RNA levels in uninduced iSLK cells, normalized to beta-actin. Error bars are mean with standard deviation, asterisks denote Student's two-tailed unpaired t-test: (siControl vs siPNUTS 1 *$p = 0.0377$), ($n = 3$). **e** RT-qPCR of T1.4 in TREx-shRNA-BCBL1 cells after induction of anti-PNUTS shRNA but without viral lytic induction, normalized to beta-actin. Error bars are mean with standard deviation, asterisks denote Student's two-tailed unpaired t-test: (shControl vs shPNUTS **$p = 0.0048$), ($n = 3$; same data as in Fig. 2i). **f** Model. PNUTS-PP1 slows pol II at the 3′ ends of most viral and host genes. A similar activity affects gene expression by promoting pausing and/or termination at the 5′ ends of KSHV ORF59 and T1.4. We hypothesize that this activity is important for additional host and viral genes (see "Discussion"). Source data are provided as a Source Data file.

ORF35 in a high-throughput interaction screen[58]. Alternatively, KSHV may use PNUTS to restrict viral gene expression as part of its carefully regulated strategy to control the latent-to-lytic switch.

PNUTS suppresses ORF59 expression from heterologous reporters and T1.4 in the viral genome. In addition, the inactivation of PNUTS leads to widespread upregulation of KSHV gene expression and more rapid virus production (Figs. 2 and 3). However, it is not clear how many of the KSHV genes that are up-regulated are direct targets of PNUTS-PP1 regulation or if indirect mechanisms of PNUTS knockdown are the primary drivers of enhanced viral replication. Given the interdependence of KSHV gene expression, it is possible that only a few genes are directly suppressed by PNUTS-PP1, but the expression of these specific transcripts is rate-limiting for viral reactivation. While ORF59 expression promotes lytic gene expression[39], our data show that its upregulation is not exclusively driving the global up-regulation of KSHV genes upon PNUTS depletion (Supplementary Fig. 2d). Alternatively, the upregulation in KSHV gene expression during latent phase after PNUTS depletion (Fig. 2c) may somehow "prime" the virus to reactivate more quickly leading to the enhanced viral gene expression and virus production at early time points after lytic reactivation. Indeed, the T1.4 transcript is generated from the oriLyt region, and its expression is necessary for lytic reactivation and oriLyt function[54,55]. It is possible that premature transcription of this essential transcript during latency lowers barriers to efficient lytic reactivation. Finally, it is possible that specific host genes are differentially expressed in the latently infected cell that modulates reactivation efficiency. Thus, we demonstrate here that PNUTS-PP1 has a robust effect on KSHV gene expression and replication, but the specific relevant targets remain unknown.

The most well characterized function of PNUTS in human genes occurs at the end of genes in the termination zone[14,25]. Importantly, PNUTS appears to maintain its termination functions for viral genes. It binds after the TES (Fig. 5d) similar to its ChIP profile in human genes, and PNUTS depletion leads to pol II readthrough downstream of viral genes (Fig. 5b, e). Nonetheless, our data indicate that the upregulation of the viral reporter and the T1.4 gene's expression upon PNUTS depletion is governed by changes in transcription at the 5′ end of the gene (Figs. 6f and 7a). Notably, PNUTS slows average pol II elongation rates within human gene bodies (~60%), just not to the extent that it does in the termination zone (~3.4-fold)[14]. Moreover, PNUTS has been reported to be a MYC co-factor at promoters[45,59]. Thus, even though human genes are not globally upregulated upon PNUTS depletion,

roles for PNUTS in promoter-proximal regions are not unprecedented, and the PNUTS mechanism on viral genes may reflect these activities.

The changes in pol II pausing that drive reporter gene upregulation could arise from any one of a number of possible mechanisms. First, PNUTS may contribute to the promoter-proximal pause that occurs on nearly all pol II transcribed genes. Indeed, PNUTS-PP1 has been implicated in dephosphorylation of the serine 5 of the CTD as well as Spt5, both of which are implicated in promoter clearance[14,20,47,60,61]. Second, pol II could be subject to PNUTS-dependent premature termination on specific viral genes. In this model, premature termination occurs on viral genes at promoter-proximal sites, not unlike the premature cleavage and polyadenylation (PCPA) events that occur in mammalian genes[62–64]. This idea is appealing in that it maintains the known activity of PNUTS in termination but suggests that it further applies near the 5′ ends of specific viral genes. Third, PNUTS may promote the clearance of stalled pol II by promoting pol II degradation. In fact, PNUTS-PP1 and its binding partner WDR82 have been implicated in promoting pol II turnover on chromatin in a proteosome-dependent manner[60], and the PNUTS-PP1 dephosphorylation substrate Spt5 has also been linked to pol II stability at promoters and processive elongation[61,65,66]. Distinguishing among these and alternative models is critical to a thorough understanding of PNUTS-PP1 mechanisms in KSHV gene expression.

Of particular interest in future studies is definition of the *cis*-acting features of ORF59, T1.4, and other KSHV genes which drive regulation by PNUTS. Preliminary analyses of human genes with shared features of short length or lack of introns did not uncover broad trends of PNUTS-dependent 5′ pausing (Supplementary Fig. 6e–g). However, previous studies have noted increases in **prom**oter **u**pstream **t**ranscripts (PROMPTs) and enhancer RNAs following loss of PNUTS[25,53]; these transcription units resemble KSHV genes in that they tend to be short and lack introns. Moreover, human intronless genes often contain *cis*-acting elements that compensate for the lack of introns in gene expression, so these may differ from viral or reporter genes that lack introns[67]. Consistent with the role for splicing and/or the presence of an intron, 5′ splice sites recruit U1 snRNP to counteract PCPA events[64]. In addition, the spliceosomal U2 snRNP promotes pause release, and PNUTS has been reported to regulate the phosphorylation of U2 snRNP component SF3B1[21,60,68]. These links between splicing and pol II pause release lead to the speculation that the lack of introns in viral genes, eRNAs, and PROMPTs drives PNUTS-responsiveness of these transcripts.

The data presented here support the conclusion that PNUTS-PP1 creates an intrinsic barrier to KSHV gene expression. Moreover, they suggest that PNUTS-PP1 suppresses gene expression by controlling elongation efficiency near the 5′-end of transcripts. We anticipate that future work will extend these observations to additional viruses, identify direct KSHV targets, and probe the detailed mechanism of PNUTS-PP1 suppression of KSHV gene expression and replication.

## Methods

### Cell culture
All cell lines were grown at 37 °C with 5% $CO_2$, with 1X penicillin-streptomycin (Sigma), and 2mM L-glutamine (Fisher). HEK293A-TOA cells were developed as previously described[7], and 293T and HCT116 cells were acquired from ATCC. The iSLK cells were a generous gift from Dr. Rolf Renne[37,38]. TREx-BCBL1 and TREx-RTA-BCBL1 cells were a generous gift from Dr. Jae Jung[40]. HCT116 cells and derivatives, 293T, HEK293A-TOA, and iSLK cells were grown in Dulbecco's modified Eagle medium (DMEM; Sigma) supplemented with 10% fetal bovine serum (FBS, Sigma). HEK293A-TOA, 293T, and iSLK cells were grown in Tet-free serum (Atlanta Biologicals). The HCT116 AAVS1-ORF59-GFP-Hyg clones were maintained in 50 μg/mL hygromycin. Inducible SLK (iSLK) cells with an integrated doxycycline-responsive KSHV ORF50 transgene and KSHV BAC16 were grown in the presence of 0.1 mg/mL G418

(Fisher), 1 µg/ml puromycin (Sigma), and 50 µg/ml hygromycin. TREx-RTA-BCBL1 cells were grown at $10^5$–$10^6$ cells/mL in RPMI 1640 (Sigma R8758) supplemented with 10% Tet-free FBS. These cells were induced into lytic reactivation by treatment with doxycycline (1 µg/ml) and sodium butyrate (NaB, 0.3 mM). TREx-shRNA-BCBL1 lines were likewise maintained in 10% Tet-free FBS and treated with doxycycline (1 µg/ml) to induce shRNA expression for 7 days prior to inducing lytic reactivation with 0.3 mM NaB.

## Plasmid construction

The plasmid hAAVS1-ORF59-GFP-Hyg was generated as follows. ORF59 was amplified from KSHV genomic DNA isolated from iSLK cells using primers NC2424 and NC2422. These, and all oligonucleotide sequences can be found in Supplementary Table 1; all plasmids are listed in Supplementary Table 2. The amplified product was inserted into the hAAVS1-CMV-GFP-Hyg vector, (a gift from Dr. Joshua Mendell), using BstBI and KpnI sites and standard molecular biology techniques[69]. The hAAVS1-ORF59-GFP-K11 plasmid was made by digestion of the hAAVS1-ORF59-GFP-Hyg plasmid with AgeI and AsiSI to remove most of the PGK promoter and part of the hygromycin resistance gene. The -2 kb KSHV K11 fragment was amplified with NC3761 and NC3762 and inserted after AgeI and AsiSI digestion using conventional cloning techniques.

To generate sgRNA expression plasmids targeting specific genes we first made a derivative of pLentiCRISPRv2, a generous gift of Dr. Feng Zhang (Addgene#52961)[70]. This plasmid, pLentiCRISPRv2TC, contains two point mutations that decrease the efficiency of PCR in the second PCR step of CRISPR screen library preparation. The mutations decrease the chances that these frequently used plasmids will contaminate subsequent unbiased library preparation. We used SOEing PCR to create pLentiCRISPRv2TC[71]. One fragment was generated by PCR using primers NC3180 and NC3181 while the second used NC3182 and NC3183; both reactions used pLentiCRISPRv2 as a template. These fragments were then used as templates for amplification using primers NC3180 and NC3183. The resulting PCR product was digested with EcoRI and XbaI and ligated to pLentiCRISPRv2 cut with the same.

To make sgRNA constructs, targeting sequences were cloned into pLentiCRISPRv2TC. For PNUTS-targeting constructs, targeting sequences from the Brunello guide RNA library were cloned into pLentiCRISPRv2TC using the BROAD Institute cloning scheme[72]. Briefly, oligonucleotides and the pLentiCRISPRv2TC plasmid were digested with BsmBI, then oligonucleotides were phosphorylated by incubation with T4 PNK (NEB M0201S) before annealing of oligonucleotide pairs and ligation into the digested plasmid. Oligonucleotides for PNUTS sgRNA #1 were NC3323/NC3324, and NC3325/NC3326 for PNUTS sgRNA #2. Non-targeting control sgRNA construct was constructed in the same way, using NC3198 and NC3199. The shRNA constructs were made by ligation of annealed hairpin oligonucleotides with overhangs compatible with AgeI/EcoRI digested pLKO-Tet-On-puro plasmid (a generous gift of Dr. Dmitri Wiederschain, Addgene #21915)[73]. PNUTS shRNA oligonucleotides were NC4006 and NC4007. Non-targeting shRNA control was a gift from Dr. Roland Friedel (Addgene #98398)[74]. All sgRNA and shRNA constructs were transformed into stable Sbtl3 E. coli.

Flag-tagged PNUTS constructs were generated by PCR amplification using a PNUTS cDNA pcDNA3.1+/C-(K)-DYK plasmid from GenScript as template (ORF clone ID: OHu03101D). Two PCR products were synthesized with overlapping silent mutations introduced to the sequence targeted by siPNUTS #1 (Supplementary Table 1). Fragment one was amplified using NC3380 and NC3632, and fragment two was amplified using NC3631 and NC3381. The siRNA target sequence was altered from GCAATAGTCAGGAGCGATA to GTAACAGCCAAGA ACGTTA. Full-length siRNA-refractory PNUTS was amplified by stitching PCR fragment one and two using NC3380 and NC3381 primers, which carry BamHI and XhoI sites, respectively. The amplified full-length PNUTS cDNA was then digested with BamHI / XhoI and cloned into an N-terminal Flag-tagged pcDNA3 plasmid. For the W401A PP1 binding mutant PNUTS, Tryptophan 401 was mutated to Alanine by introducing a TGG to GCT codon change using overlapping PCR fragment SOEing as described above, amplified from the siRNA refractory construct. Fragment one was amplified with primers NC3380 and NC3655, and fragment two using NC3654 and NC3381. Amplified fragments were subsequently used for PCR with NC3380 and NC3381 and the resulting product was cloned into pcDNA3-Flag as above.

## Generation of GFP-reporter lines

To integrate the ORF59-GFP reporter into the AAVS1 locus, HCT116 cells were co-transfected in a 6-well plate with 0.2 µg AAVS1 1 L TALEN, 0.2 µg AAVS1 1R TALEN, and 1.6 µg hAAVS1-ORF59-GFP-Hyg[70]. The next day cells were split to 10 cm plates, and hygromycin was added to 200 µg/ml. Cells were selected for a total of 10 days. Fluorescence-activated cell sorting (FACS) was used to select single cells with low to moderate GFP signal for expansion. Clonal cell lines were further screened by two methods. First, we assayed for increased GFP expression upon KSHV ORF57 expression. Second, we analyzed GFP levels after depletion of nuclear RNA decay factors[44]. Those clones that increased GFP in both these conditions were selected for further experiments. To generate the ORF59-GFP-K11 reporter line, stable cell clones were generated using the same technique with the hAAVS1-ORF59-GFP-K11 donor plasmid, except no hygromycin selection was performed.

## Transfection and siRNA knockdown

All siRNA knockdowns were performed on cells at 60–80% confluency with 30 nM siRNA (Silencer Select; ThermoFisher) using RNAiMax transfection reagent (Invitrogen) according to the manufacturer's protocol. For co-depletion assays, 60 nM siRNA was used overall, with 30 nM for each target. Catalog numbers for specific siRNAs are listed in Supplementary Table 1. The day after siRNA transfections, cells were split ~1:4 to new plates. For iSLK knockdown and reactivation experiments, iSLK cells were induced with 1 µg/ml doxycycline and 1 mM NaB two days after siRNA transfection. For plasmid transfection assays, cells were transfected with plasmid DNA two days after siRNA treatment using Transit293 transfection reagent (Mirus) and harvested 24 h later.

## Western blotting

Total protein was harvested by direct lysis with SDS protein buffer (100 mM Tris-HCl pH 6.8; 20% Glycerol; 4% SDS; 2% 2-mercaptoethanol; 0.1% bromophenol blue) and resolved by SDS-PAGE by standard western blotting protocols. All antibodies and dilutions are listed in Supplementary Table 2. Imaging of western blots was done with infrared detection with Odyssey Fc and signal was quantified by ImageStudio (v5.2) software (Li-Cor Biosciences).

## Northern blotting

Total RNA was harvested using TRI reagent (Molecular Research Center, Inc), and processed according to the manufacturer's protocol. Northern blots were probed with radiolabeled RNA probes produced using templates made by plasmid digestion or PCR products with T7 or SP6 RNA polymerase promoter sequences. Briefly, 3 µg total RNA per lane was resolved on 1.2% formaldehyde-agarose gels and transferred to nylon membrane (Hybond N+, GE Healthcare). Probes were in vitro transcribed with 25 uCi of $\alpha$-$^{32}$P-UTP (800 Ci/mmol); 0.5 mM ATP, CTP, GTP, and 0.1 mM UTP; 40 mM Tris pH-7.5; 6 mM MgC $l_2$; 10 mM DTT, 200 ng DNA template, RNAsin Plus (Promega), and T7 or SP6 polymerase. Probes were hybridized overnight to membranes in Church's hybridization buffer (7% SDS, 15% formamide, 1 mM EDTA, 1% bovine serum albumin, 200 mM phosphate buffer) at 65 °C[75]. Blots were then

washed before imaging in 0.1% SDS washes with 2×, 0.5×, and 0.1× SSC (150 mM NaCl, 15 mM sodium citrate). For the readthrough transcription blots, probes were synthesized with 0.5 mM UTTPαS (TriLink Biotechnologies). After imaging, these blots were stripped for reprobing by incubation at 65 °C in strip buffer (7% SDS, 50% formamide, 20 mM phosphate buffer, and ~6 mM $I_2$). All blots were imaged with a Typhoon FLA 9500 Phosphorimager (GE Healthcare), and bands quantified using ImageQuant v5.2.

For polyA-selected northern blots, 40 μL of 50% slurry Sera-Mag oligo(dT)-coated magnetic particles (GE Healthcare Lifesciences) was washed and equilibrated in 1×SSC with 0.1% SDS and 80 μg total RNA was added. The samples were then nutated for 20 min at RT, washed three times in 0.5× SSC with 0.1% SDS before elution of the bound RNA with water for 5 min, once at room temperature (RT) and again at 65 °C. Combined eluted RNA was purified with phenol:chloroform:isoamyl alcohol 25:24:1 (PCA) extraction, and an additional chloroform extraction before ethanol precipitation.

## CRISPR screen

Genome-wide CRISPR screen was performed as described in detail in Scarborough et al. except transduced reporter cells were maintained in normal media plus puromycin and sorted on day 7 after lentiviral transduction for the top 8% highest GFP-expressing cells[76]. Briefly, cells were plated at ~4 × $10^5$ cells/well on sixteen 6-well plates and transduced with lentivirus carrying the human Brunello CRISPR knockout pooled library (a gift from Drs. David Root and John Doench; Addgene #73178)[35]. Lentivirus titer was adjusted to yield ~20% infection of cells to avoid co-infection with more than one guide per cell. Overall coverage for the library was 100× at infection. Two days after infection, cells were selected with 1 μg/mL puromycin and expanded as necessary. Seven days post infection, cells were sorted for the highest ~8% GFP expressing cells using a FACSAria Fusion Cell Sorter. Minimum coverage for all three biological replicates calculated at the cell sorting was 200× (calculation details described in Scarborough et al., 2021)[76]. Sorted and unsorted input cell samples were harvested, and sequencing libraries were made using a two-step PCR variation of the BROAD Institute protocol[72]. Libraries were single-end sequenced configured to 75 bp on an Illumina NextSeq 500 with an average of 15.7 million reads per sample. Analysis was performed with MAGeCK-VISPR workflow, with batch correction of three biological replicate screens[77]. The CRISPR data are posted under GEO accession number GSE200991, under a super-series with all following datasets with the GEO accession number GSE201046.

## RT-qPCR

RNA was harvested with TRI reagent according to the manufacturer's protocol. Purified RNA was treated with RQ1 DNase (Promega) and cDNA was synthesized with murine leukemia virus (MuLV) reverse transcriptase (New England Biolabs) with oligo(dT)20 priming. All real-time PCR mixes performed with iTaq Universal SYBR green Supermix (Bio-Rad) and analyzed with QuantStudio (v1.5.2). Gene-specific primer sequences can be found in Supplementary Table 1.

## RNA-Seq

iSLK cells were transfected with siRNA and induced into lytic replication. Cells were harvested in TRI Reagent at induction (0hpi), after 8, 24, and 48 h of lytic reactivation. One microgram of total RNA per sample was processed with KAPA Stranded RNA-Seq kit with RiboErase (KAPA Biosystems) according to the manufacturer's protocol. Strand-specific single-end RNA sequencing was performed on an Illumina HiSeq 2500. Reads with Phred quality scores less than 20 and less than 35 bp after trimming were removed from further analysis using Trim Galore (v0.4.1)[78]. Quality-filtered reads were then aligned to the human reference genome hg19, with KSHV GQ994935.1 as an extra chromosome, using the HISAT (v2.0.1)[79] aligner with default settings.

Duplicates were marked but retained using Sambamba (v0.6.6)[80]. Aligned reads were quantified using featureCounts (v1.4.6)[81] per gene ID against GENCODE GRCh37[82]. Differential gene expression analysis was performed using the R package DESeq2 (v1.6.3)[83]. Cutoff values of absolute fold change greater than 2 and FDR < 0.1 were used to select for differentially expressed genes between sample group comparisons. RNA-Sequencing data are posted under GEO accession number GSE201000.

## Lentivirus production

All sgRNA and shRNA lentiviruses were generated in the same fashion. Approximately 50–70% confluent 293T cells were transfected with the target pLentiCRISPRv2 or pLKO-Tet-On derived plasmid along with packaging plasmids psPAX2 and psPMD.2 (gifts from Didier Trono, Addgene #12260, #12259). Target plasmid, psPAX2, and psPMD.2 were transfected at 5:3:2 ratios, respectively. Media was harvested two days after transfection, filtered with 0.45 μM filters and supplemented with 20 mM HEPES. Lentivirus particles were concentrated ~4–6 fold by centrifugation at 1000 × g at 4 °C with Amicon Ultra-15 centrifugal filters (Merck).

## Generation of sgRNA TREx-BCBL1 and TREx-shRNA-BCBL1 cell lines

Non-targeting control or sg/shPNUTS lentivirus particles were mixed with 2 × $10^6$ TREx-RTA-BCBL1 or TREx-BCBL1 cells supplemented with 20 mM HEPES and 8 μg/ml polybrene (Sigma) and centrifuged at 800 × g for 45 min at 32 °C, then incubated overnight. Forty-eight hours post-transduction, cells were selected by puromycin addition (3 μg/ml). To induce expression of shRNAs, cells were passaged at 2 × $10^5$ cells/mL with 1 μg/ml doxycycline. After 7 days in doxycycline, cells were induced into lytic viral replication with 0.3 mM NaB, and total RNA and protein harvested after 24, 48, and 72 hpi. Viral supernatant transfer assays performed as described below, except the recipient cells were assayed by harvesting total RNA followed by RT-qPCR analysis of latent KSHV gene expression.

## vDNA timecourse

Total DNA was harvested with DNAzol Reagent (Fisher) using the manufacturer's protocol. iSLK cells were treated with siRNA and induced into lytic reactivation as described. Purified DNA was assayed by qPCR using 5 ng total DNA per reaction as template using primers NC3030/NC3031 for KSHV ORF58, and host DNA was assayed using 7SK primers NC1164/NC1165.

## KSHV infection assays

For KSHV infection assays, supernatants from iSLK and TREx-shRNA-BCBL1 cells were harvested from lytically induced cells. Supernatants were centrifuged for 5 min at 1000 × g, filtered with 0.45 um filters, and supplemented with 20 mM HEPES before flash-freezing in liquid nitrogen. Supernatants from iSLK cells were pre-treated with Benzonase for 30 min at 37 °C before infection (Sigma). For spinfection, supernatants were mixed 1:1 with regular media and 8 μg/ml polybrene and added to ~50–70% confluent HCT116 cells. After centrifugation for 45 min at 800 × g and 32 °C, plates were incubated at 37 °C for 2hrs before infection media was removed and replaced with fresh media. Twenty-four hrs later, HCT116 cells were harvested with TRI reagent for TREx-shRNA-BCBL1 experiments. For iSLK experiments, cells were trypsinized and pelleted at 800 × g for 5 min at 4 °C, resuspended in 1% formaldehyde in PBS, and fixed overnight at 4 °C for flow cytometry.

## Flow cytometry

Formaldehyde fixed samples were pelleted and resuspended in 1× PBS with 3% FBS and analyzed with a Stratedigm S1000 flow cytometer using CellCapture (v3.1) software (Stratedigm, Inc). Data analysis was performed with FlowJo (v 9.9.5) (FlowJo LLC).

## ChIP

For ChIP assays, cells were fixed with 0.5% formaldehyde at RT for 10 min then quenched with 150 mM glycine at RT, 5 min. Cells were harvested by scraping in cold 1× PBS and pelleted at 1000 × $g$, 5 min at 4 °C. Harvested cells were lysed for 30 min at 4 °C in Farnham lysis buffer (5 mM PIPES, 85 mM KCl, 0.5% NP-40, 1 mM PMSF, 1× protease inhibitor cocktail (PIC)). Samples were Dounce homogenized and then 20 × 10⁶ nuclei per sample were collected by centrifugation (1000 × $g$, 5 min at 4 °C) and resuspended to 25 × 10⁶ cells/mL in Szak's RIPA Buffer (50 mM Tris-HCl pH 8, 1% NP-40, 150 mM NaCl, 0.5% Deoxycholate, 0.1% SDS, 2.5 mM EDTA, 1 mM PMSF, and 1× PIC). Chromatin was fragmented to an average size distribution of ~100–300 base pairs by sonication for 25 min at 4 °C (30 s ON, 30 s OFF at 50% Amp) using a QSonica Q800 sonicator. 50 ng of *Drosophila* spike-in chromatin was added to each sample (Active Motif Cat. #53083). Fragmented chromatin was pre-cleared by 50 μl of Dynabeads Protein A (Invitrogen) by nutation at 4 °C for 1 h. In parallel, 100 μl per sample of 50% slurry Dynabeads Protein A were conjugated to 5 μg anti-RPB3 antibody (Cat. #ABE999, Millipore) and 2.5 μg antibody for the *Drosophila*-specific histone variant H2Av (Active Motif Cat. No 61686) by nutation at 4 °C for 1 hr in 1× PBS-Tween (0.05%). Antibody-coated beads were then washed and equilibrated in Szak's RIPA buffer before blocking with 5% BSA in Szak's RIPA Buffer for 1 h at 4 °C. Input samples (5%) were taken from pre-cleared chromatin, then the remainder added to the blocked antibody coated beads, and the ChIP was performed overnight at 4 °C. All ChIP washes were performed twice, for 5 min at 4 °C, with nutation. ChIP wash solutions were as follows: Szak's RIPA buffer, low salt buffer (0.1% SDS, 1% NP-40, 2 mM EDTA, 20 mM Tris-HCl pH 8, 150 mM NaCl), high salt buffer (as in low salt buffer, except with 500 mM NaCl), LiCl buffer (250 mM LiCl, 1% NP-40, 1% Deoxycholate, 1 mM EDTA, 20 mM Tris-HCl pH 8), and TE buffer (10 mM Tris-HCl pH 8, 1 mM EDTA). Immunoprecipitated chromatin was eluted from the beads for 30 min at 65 °C in elution buffer (100 mM NaHCO₃, 1% SDS), then crosslinks were reversed overnight at 65 °C in decrosslinking buffer (500 mM NaCl, 2 mM EDTA, 20 mM Tris-HCl pH 6.8, 0.5 mg/mL Proteinase K). Input samples were treated with the elution and crosslink reversal steps in parallel. Finally, input and ChIP DNA was processed with Zymo research ChIP DNA clean and concentrator kit (Zymo Cat. #D5201) before validation by qPCR. For ChIP-seq library preparations, 4 ng of ChIP or input DNA was processed using the KAPA Hyper Prep Kit according to the manufacturer's protocol (Roche KK8504). Size-selected and amplified libraries were submitted for next-generation sequencing with Illumina HiSeq 2500, configured to 75 bp single-end reads.

## ChIP-Seq analysis

For ChIP-Seq in the reporter HCT116 cell line, chimera genome assembly and genome annotation were constructed by integrating ORF59-GFP reporter into human genome at position chr19:55,115,766–55,121,825. Raw Fastq reads were checked for data quality using FastQC/0.11.9 and filtered for low-quality base calls (Phred score <20) and adapter contaminations using trimgalore/0.6.4. Reads were then aligned to a combined genome constituted of human (GRCh38 with reporter), and *Drosophila* (dm6) using Bowtie2/2.3.3 with default parameters. The aligned reads were filtered for mapping quality (MAPQ 30) and PCR duplicates by SAMtools/1.6 and Picard/1.127 respectively. Uniquely aligned reads corresponding to *Drosophila* (spike-in) were subsequently separated from the human prior to calculating size factors for spike-in normalization using DESeq2 (v1.6.3). The size factors were then applied to reads aligned to human genome using bamCoverage of deeptools suit (2.5.0.1). The ChIP-seq data for two biological replicates for each condition were highly correlated (Pearsons' correlation ≥ 0.98). Input normalized bigWig tracks were produced using bamCompare for metagene analysis and for visualization in the Integrative Genomics Viewer (IGV).

Pausing indices were determined by calculating the ratio of read coverage in the promoter-proximal region (−50/+300 bp relative to transcription start site, TSS) and promoter-distal region (+301 bp to +300 bp past TES) as described previously[84]. Non-active genes (with low read coverage in the promoter-proximal region) were excluded from the analysis of PI of protein-coding endogenous genes. For the genes with multiple annotated isoforms, only the isoform with the strongest signal at promoter-proximal region was selected. Genes that were less than 1 kb long or within 10 kb of nearby genes were excluded from the PI and metagene analysis. A custom Perl script was used to extract genomic coordinates of intronless and lincRNA genes from GENCODE GTF (v27). ChIP-Sequencing data in HCT116 cells are posted under GEO accession number GSE200992. ChIP-Seq data analysis for iSLK cells was performed as above except samples were normalized to per million mapped reads (RPM) and mapped to KSHV GQ994935.1 as an extra chromosome. ChIP-Seq data from iSLK cells are posted under GEO accession number GSE215345.

## 4SU capture of newly made transcripts

Cells were incubated with 500 μM 4-Thiouridine (4SU, TriLink Biotechnologies) for 10 min at 37 °C, then the RNA was harvested in TRI reagent and treated with RQ1 DNase. To biotinylate the 4SU-labeled RNA, 80 μg per sample of purified RNA was incubated for 3 h at 25 °C with 0.2 mg/mL Biotin-HPDP (Thermo Fisher) in 10 mM Tris-HCl pH 7.5, 1 mM EDTA, and 0.1% SDS. After biotinylation, RNA was extracted twice with chloroform and precipitated with ethanol and 1 M ammonium acetate. To precipitate biotinylated RNA, 20 μl of a 50% slurry per sample of Dynal MyOne Streptavidin T1 beads (Thermo Fisher) were washed and equilibrated in MPG 1:10-I (100 mM NaCl, 1 mM EDTA, 10 mM Tris-HCl pH 7.5, 0.1% Igepal). Beads were blocked for 1 h at room temperature in MPG 1:10-I supplemented with 0.1 μg/μL poly(A) RNA (Sigma), 0.1 μg/μL cRNA, 0.1 μg/μL sheared salmon sperm DNA (Sigma) and 0.1% SDS. An aliquot (5%) of the resuspended biotinylated RNA was saved as input, and the remainder was mixed with the blocked beads and nutated at room temperature for 1 h. Beads were washed eight times: MPG1:10-I, MPG1:10 (no Igepal) at 55 °C, twice with MPG-I (1 M NaCl, 10 mM EDTA, 100 mM Tris-HCl pH 7.5, 0.1% Igepal), MPG1:10-I, MPG-I no salt (10 mM EDTA, 100 mM Tris-HCl pH 7.5, 0.1% Igepal), MPG1:10-I. RNA was eluted from the beads twice in MPG 1:10-I with 5% beta-mercaptoethanol for 5 min at room temperature. Combined elutions were PCA and chloroform extracted before ethanol precipitation. Total precipitated sample was then processed for RT-qPCR.

## eCLIP

The eCLIP assays were performed by Eclipse Bioinnovations following the protocol detailed in van Nostrand et al.[52]. For these assays, the anti-PNUTS antibody A300–439A-T (Bethyl Laboratories) was used, and two independent biological replicates were performed. An IgG control antibody was tested in parallel, but the yield from this was low, so all subsequent comparisons were done relative to the size-matched input control. Sequences were processed and mapped using the pipeline described in van Nostrand et al. except the reads were simultaneously mapped to GRCh37/hg19, and the KSHV genome GQ994935.1. Metagene analyses were anchored to the 41 unique KSHV RNA transcription end sites found in Supplementary Table 3[32,33]. Metagenes were made using aligned bam files obtained from Eclipse Bioinnovations. Data analyses was performed with the deeptools suite[85]. Default options with bamCompare was used to create bigwig files from sample versus control bamfiles. Then, computeMatrix was used to calculate metagene signal across regions of interest. Finally, plotProfile was used with default options to produce metagene profile plots. The eCLIP data are posted under GEO accession number GSE201045.

## Analysis and statistics

RNA-Sequencing, eCLIP, and ChIP-Seq profiles were visualized with the Integrative Genomics Viewer (IGV v2.9.2). Heatmaps, bar graphs, line graphs, and box/violin plots were all made using GraphPad Prism (v9) software. Bar graphs represent a minimum of three biological replicates with mean value displayed and error bars denoting standard deviations. Asterisks denote $p$-values calculated in GraphPad Prism using unpaired two-tailed t-tests: $*P < 0.05$, $**P \leq 0.01$, $***P \leq 0.001$, n.s., not significant.

## Reporting summary

Further information on research design is available in the Nature Portfolio Reporting Summary linked to this article.

## Data availability

Data generated in support of this study are provided in data figures and supplemental files. CRISPR screen sequencing datasets, eCLIP, RNA-Seq, and ChIP-Seq datasets are available in the GEO super-series GSE201046. Reference genomes used are publicly available at NCBI with the following accession numbers: hg19, GRCh38, KSHV GQ994935.1. Source data are provided with this paper.

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

## Acknowledgements

The authors thank Drs. David Root, John Doench, Joan Steitz, Feng Zhang, Joshua Mendell, Didier Trono, Dmitri Wiederschain, and Roland Friedel for reagents. We thank members of the Conrad lab for helpful comments on the manuscript. This work was supported by National Institutes of Health National Institutes of Allergy and Infectious Disease grants R01AI114362 (to I.D.), R01AI123165 (to N.K.C.), and R01AI153175 (to N.K.C.).

## Author contributions

N.K.C. and A.M.D. conceived and designed the experiments, interpreted results, and wrote the manuscript. A.M.D. performed the experiments. A.S. performed all bioinformatic analyses of the ChIP-Sequencing datasets and wrote the respective methods information. J.C.R. contributed cloning and cell line construction. S.D.B. performed bioinformatic analyses of RNA-Sequencing and eCLIP datasets. A.G. performed bioinformatic analyses of CRISPR screen datasets. O.V.H. contributed cloning construction and assisted in the production of Fig. 4 data. A.M.S. assisted in the completion of the CRISPR screens. I.D. assisted in the conception and interpretation of the ChIP-Sequencing experiments. All authors provided critical review of the manuscript.

## Competing interests

The authors declare no competing interests.
