## [Peer Review File · Nature Communications]

Reviewer comments, first round review –

Reviewer #1 (Remarks to the Author):

The authors investigate cellular factors that may suppress KSHV transcription/elongation by using a reporter KSHV gene and a CRISPR screen. They convincingly identify PNUTS, a PP1 targeting protein involved in transcription termination by slowing pol II elongation, as inhibiting KSHV ORF59 mRNA accumulation, both by CRISPR and siRNA transfection. In experiments with KSHV infected cells, they demonstrate that PNUTS KD leads to global increases in KSHV lytic gene expression and viral replication, as well as an earlier onset of such lytic replication. In the reporter assays they also demonstrate that the activity of PNUTS requires the phosphatase activity of PP1, indicating that the mechanism of suppression is based on its known activity in slowing pol II. Based on ChIP experiments with the ORF59 reporter gene, they find that PNUTS depletion leads to increased gene body occupancy by pol II and that promoter-proximal pausing is reduced. Such an effect was not seen with cell genes globally.

This paper makes several very interesting observations and substantiate a potential role for PNUTS in regulation of KSHV gene expression. The experiments are very carefully done with appropriate controls and the data are convincing. This is significant to the KSHV field. The authors state that due to the gene structure and organization of the viral genome, it was not possible to measure pol II occupancy along the gene. This leads to the support for the PNUTS pausing model in KSHV repression to be primarily based on its effect on the ORF59 reporter. While they do show that many lytic genes are upregulated by PNUTS KD, it raises the question of how exactly PNUTS has this broad repressive effect. Is one to assume that all the KSHV genes have this elongation block? Looking at their transcriptome in Figure 2, there appear to be at least a few genes that are not upregulated by PNUTS KD. Is there some structure or sequence that makes ORF59 and some genes particularly susceptible to pausing?

The broad effect on gene expression of PNUTS KD could also be indirect and not directly due to an effect on all the lytic genes. ORF59 has been shown to modulate histone arginine methylation of the viral genome to promote viral reactivation. PNUTS repression of ORF59 alone could thus lead to an apparent global effect of PNUTS on KSHV lytic genes even if the proposed pausing mechanism does not apply to them. A similar concern is raised by the fact that ORF59 is required for DNA replication, formation of replication/transcription factories and thus late gene expression.

The authors also examine host cell genes to see if particular types of genes that may resemble KSHV genes show this effect of PNUTS KD on the pausing index. They should identify cellular genes where this does indeed happen and see if these have some identifiable characteristics. The bioinformatic analyses they have done should allow this to be easily performed. Similarly, an examination of such cellular genes which are upregulated or downregulated by PNUTS KD to look at their function needs to be done. Such an ontologic analysis could reveal whether PNUTS KD acts by mechanisms besides elongation block to inhibit KSHV gene expression.

Based on the above issues, an examination of at least one other KSHV gene to demonstrate the promoter proximal pausing block by PNUTS would seem to be required to fully validate the model presented in this very intriguing paper.

Reviewer #2 (Remarks to the Author):

This study by Devlin et al. used a CRISPR screen to identify PNUTS as a negative regulator of KSHV ORF59 reporter expression. The authors go on to show that PNUTS limits virus production and perform mechanistic experiments that support roles of PNUTS in transcription termination (the best known role of PNUTS in cellular gene expression) and, in the case of ORF59, 5' POL II occupancy. This topic is interesting, since KSHV genes present several obstacles for typical mRNA transcription, processing, and export by cellular machinery. Overall, this is a carefully executed

study and a well written manuscript. The data that are presented are well analyzed and clear. Conclusions are supported by the data. One potential point of criticism is an over-reliance on reporter studies. While reporter studies tend to give clear results, they may not capture the full biology of the infection context. Specifically, the promoter proximal effect is only demonstrated for an ORF59 mRNA that is expressed from a heterologous CMV promoter. Minimally, the authors should be cautious to not overinterpret these findings, e.g. in abstract line 33. While the PNUS knock-down phenotype is characterized, the specific viral features the underly this phenotype remain unknown. There are also some other missed opportunities: e.g. it could easily have been established if ORF59 overexpression would phenocopy PNUTS knockdown and/or assessed/discussed whether ORF57 is involved in facilitating ORF59 expression in the presence of PNUTS (the intial portion of the manuscript almost sets up this question). In sum, this study is a carefully executed study of sufficient interest for publication largely as is.

I only have minor requests for clarification:

PPR1R10/PNUTS is a broadly essential gene – the authors should at least mention broader toxicity of PNUTS KO, since it complicates the interpretation of the presented knock-down studies due to toxicity and could explain less pronounced phenotypes later into reactivation. Was toxicity observed outside of BCBL-1?

Line 171: the authors discuss expression of ORF50 after PNUTS knockdown, i.e. exclude a premature induction phenotype, but this experiment was performed under conditions of ectopic ORF50 expression. It is therefore not clear whether this interpretation would hold up in a more physiological context. It should also be explained how ectopic and viral ORF50 reads are distinguished in this experiment.

Details of TALEN-facilitated gene insertion should be cited or described (lines 524/525).

Reviewer #3 (Remarks to the Author):

In their manuscript entitled "The PNUTS-protein phosphatase 1 complex acts as an intrinsic barrier to KSHV gene expression and replication", the authors identify the PNUTS-PP1 complex as a critical suppressor of lytic KSHV gene expression using a genome-wide CRISPR screen. They show that mechanistically PNUTS requires PP1 interaction. Surprisingly, PNUTS was also found to represses productive elongation at the 5'-end of the KSHV reporter gene.

The manuscript is well written and easy to follow. Besides the identification of PNUTS, the CRISPR screen provides exciting data on the involvement of other cellular genes with known roles in transcription. While the exact molecular mechanism remains unclear the discussion provides an excellent overview on the putative mechanism. The paper will be of importance for many groups in the field of transcription.

I have no major comments. The only minor aspect I found a bit irritating when I first read the paper was "PNUTS-protein phosphatase 1 (PP1) complex" in the abstract as PNUTS was abbreviated but PP1 was not. I would suggest to replace this by "PNUTS-PP1 complex" and only explain this in the main manuscript.

We thank each of the reviewers for their overall very positive comments regarding our initial submission. In the revised version, we have addressed each of those concerns and have included a new figure that deals with the primary concern of Reviewers #1 and #2 regarding our reliance on reporter genes. We think these changes have improved the rigor of the work and bolster our conclusions. Detailed responses are listed below with the original reviewers' comments in blue italics and our responses in black regular font. In the revised manuscript, all text changes are in blue font.

Reviewer #1 (Remarks to the Author):

1.1. This paper makes several very interesting observations and substantiate a potential role for PNUTS in regulation of KSHV gene expression. The experiments are very carefully done with appropriate controls and the data are convincing. This is significant to the KSHV field. The authors state that due to the gene structure and organization of the viral genome, it was not possible to measure pol II occupancy along the gene. This leads to the support for the PNUTS pausing model in KSHV repression to be primarily based on its effect on the ORF59 reporter.

1.2. Based on the above issues, an examination of at least one other KSHV gene to demonstrate the promoter proximal pausing block by PNUTS would seem to be required to fully validate the model presented in this very intriguing paper.

- We have combined Reviewer #1's first and final comments as they both relate to the concern that our mechanistic interpretations were based almost completely on a reporter gene. The same issue was also the primary concern of Reviewer #2.
- While our initial submission showed that PNUTS affected viral gene expression in reporters and in the context of KSHV reactivation, the reviewers rightly point out that our mechanistic conclusions were based solely on the reporter. As such, we did not demonstrate that this mechanism is operative in the context of the virus. Unfortunately, the overlapping transcription units and high levels of pol II on the viral genome during KSHV lytic reactivation preclude the use of standard pol II ChIP-seq during lytic phase. However, there's significantly less transcription genome-wide prior to induction and some viral genes are upregulated in the uninduced state upon PNUTS depletion. Therefore, we performed ChIP-seq on uninduced cells \pm PNUTS and have included these data in the revised manuscript (Fig 7).
- Excitingly, we report that the viral 1.4 kb transcript (T1.4) shows a pol II pattern consistent with our reporter. That is, there is similar loading at the promoter \pm PNUTS, but there is considerably more pol II downstream after PNUTS knockdown. This is reflected in the pausing index and in the RNA levels of the transcript. By showing similar decreases in pausing index between our reporter and an bona fide viral gene produced from the viral genome, we think these data

strongly support our conclusion that a subset of viral genes is repressed by PNUTS activity in promoter-proximal regions.

1.3. While they do show that many lytic genes are upregulated by PNUTS KD, it raises the question of how exactly PNUTS has this broad repressive effect. Is one to assume that all the KSHV genes have this elongation block? Looking at their transcriptome in Figure 2, there appear to be at least a few genes that are not upregulated by PNUTS KD. Is there some structure or sequence that makes ORF59 and some genes particularly susceptible to pausing?

- These are excellent points. To be clear, we do not think that all KSHV genes have this elongation block. In fact, our new data show that this is not the case for at least some of the latent genes (Extended data Fig. 7b). As the reviewer suggests, we think that there are cis-acting determinants that make a particular gene more or less subject to PNUTS-mediated elongation block. We hypothesize that this may be due to the presence of an intron/splicing, but this is not absolute because GFP reporter alone (i.e., without ORF59 fusion) is not PNUTS-sensitive. As such, factors like GC-richness or other unknown parameters may be important. The amount of work required to rigorously define these determinants will encompass an entire paper(s), but we have integrated this idea into the Discussion to clarify that more work needs to be done (lines 472-486).
- Perhaps more importantly, this point combined with a similar suggestion from Reviewer #2 prompted our realization that our language was sometimes overly general (point 2.2 below). Therefore, we now refer to the process as affecting the reporter, “a subset of viral genes”, or “some viral genes” rather than using the more all-encompassing term “viral genes” (e.g., lines 117, 110, 399, et al.)

1.4. The broad effect on gene expression of PNUTS KD could also be indirect and not directly due to an effect on all the lytic genes. ORF59 has been shown to modulate histone arginine methylation of the viral genome to promote viral reactivation. PNUTS repression of ORF59 alone could thus lead to an apparent global effect of ORF59 on KSHV lytic genes even if the proposed pausing mechanism does not apply to them. A similar concern is raised by the fact that ORF59 is required for KSHV replication, formation of replication/transcription factories and thus late gene expression.

- We agree with the reviewer that due to the interdependency of viral gene expression, release of repression of a few genes could be driving the global responses we see upon PNUTS knockdown. Since we don't know what fraction of PNUTS genes have this PNUTS-dependent pause (see above), it remains possible that the global effect is due to the activity of a few or even one single gene(s). This is now addressed more thoroughly in the Discussion (lines 421-441).

- As pointed out by the reviewer, ORF59 activities could contribute to the enhanced viral reactivation efficiency observed after PNUTS knockdown. To test this more directly, we examined expression of viral genes after knocking down PNUTS upon co-depletion of ORF59. If ORF59 is the only factor driving enhanced gene expression upon PNUTS depletion, reducing its expression to normal levels in PNUTS-depleted cells will abrogate the effect of PNUTS knockdown on other viral genes. In contrast to this prediction, other viral genes remained PNUTS sensitive in the absence of ORF59. Importantly, this experiment does not prove that ORF59 has no effect on viral gene expression after PNUTS depletion, but it does rule out that ORF59 is the sole driving effector of the viral gene up-regulation in response to PNUTS loss. These data are now included in Extended Data Fig 2d.
- The upregulation of the T1.4 after PNUTS knockdown could be helping drive more efficient lytic replication. T1.4 is transcribed over the oriLyt region, and, in principle, its overexpression could “prime” the virus for more efficient lytic reactivation. Our ongoing studies will focus on determining which subset of viral genes are direct targets of PNUTS and also which genes (if any) are necessary/sufficient to drive more efficient lytic reactivation. These studies will take years to complete, but we now discuss these points in more depth (lines 428-441)

1.5. The authors also examine host cell genes to see if particular types of genes that may resemble KSHV genes show this effect of PNUTS KD on the pausing index. They should identify cellular genes where this does indeed happen and see if these have some identifiable characteristics. The bioinformatic analyses they have done should allow this to be easily performed.

- We agree with the reviewer that PNUTS likely plays a role at the promoter-proximal regions of a subset of human genes, so we examined pausing index for human genes. Unfortunately, the analysis is not as easy as anticipated. The reason is that PNUTS knockdown induces a transcription termination defect, and this leads to “read-in” of adjacent genes, even at long genomic distances. The

resulting “read-in” ChIP-seq reads are indistinguishable from bona fide elongation events. Thus, we observed many false positives. As an example, we included a genome browser screen shot of the RABEPK gene, which was bioinformatically identified as a human gene with a decreased pausing index after PNUTS knockdown. Visual inspection of the locus makes it clear that the increased signal in the gene body of RABEPK in the siPNUTS samples is due to readthrough from the downstream convergent HSPA5 gene. Due to these complications, we have not included pausing index data for human genes as it may be misleading.

1.6. Similarly, an examination of such cellular genes which are upregulated or downregulated by PNUTS KD to look at their function needs to be done. Such an ontologic analysis could reveal whether PNUTS KD acts by mechanisms besides elongation block to inhibit KSHV gene expression.

- We performed GO analysis with two different pipelines, GOEnrichment and GOrilla. For the latter, only two general categories—multicellular organismal process and regulation of multicellular organismal process—reached a FDR<0.05. GOEnrichment pipeline did not show any enrichments for molecular function or biological process. Thus, these GO analyses are not particularly informative, so we have not reported them here. The lack of clear GO category enrichment is likely related to the general nature of PNUTS activity on termination of many genes combined with the non-specific effects of pol II readthrough on gene expression (e.g., promoter occlusion).

Reviewer #2 (Remarks to the Author):

This study by Devlin et al. used a CRISPR screen to identify PNUTS as a negative regulator of KSHV ORF59 reporter expression. The authors go on to show that PNUTS limits virus production and perform mechanistic experiments that support roles of PNUTS in transcription termination (the best known role of PNUTS in cellular gene expression) and, in the case of ORF59, 5' POL II occupancy. This topic is interesting, since KSHV genes present several obstacles for typical mRNA transcription, processing, and export by cellular machinery. Overall, this is a carefully executed study and a well written manuscript. The data that are presented are well analyzed and clear. Conclusions are supported by the data.

2.1. One potential point of criticism is an over-reliance on reporter studies. While reporter studies tend to give clear results, they may not capture the full biology of the infection context. Specifically, the promoter proximal effect is only demonstrated for an ORF59 mRNA that is expressed from a heterologous CMV promoter. While the PNUTS knock-down phenotype is characterized, the specific viral features the underly this phenotype remain unknown.

- We have addressed this in detail above in response to Reviewer #1 points 1.1-1.3. To summarize, we now have report that a viral gene (T1.4) is affected

promoter-proximally by PNUTS. In addition, we have changed the language throughout the paper to more carefully reflect that we think PNUTS affects a subset of genes (i.e., not necessarily all viral genes).

2.2. There are also some other missed opportunities: e.g. it could easily have been established if ORF59 overexpression would phenocopy PNUTS knockdown and/or assessed/discussed whether ORF57 is involved in facilitating ORF59 expression in the presence of PNUTS (the initial portion of the manuscript almost sets up this question). In sum, this study is a carefully executed study of sufficient interest for publication largely as is.

- We examined the effects of ORF57 overexpression \pm PNUTS depletion in transient reporter assays. However, the results were highly variable and different normalization schemes led to different interpretations. We suspect that this results from the multifunctional nature of ORF57. Because of these difficulties, we have left these data out of the revised manuscript.
- See response to Reviewer #1 above, point 1.4, regarding the whether ORF59 is an effector of the PNUTS phenotype.

I only have minor requests for clarification:

2.3. PPPR1R10/PNUTS is a broadly essential gene – the authors should at least mention broader toxicity of PNUTS KO, since it complicates the interpretation of the presented knock-down studies due to toxicity and could explain less pronounced phenotypes later into reactivation. Was toxicity observed outside of BCBL-1?

- By visual inspection, the cells seemed relatively healthy 3-day knockdowns of PNUTS in uninduced iSLK, HCT116, and HEK293-ATOA lines. That is, we observe limited cell rounding and only slight slowdowns of cell division at this time point. We mention this in the revised version on page 9 (lines 191-194). We do observe a strong synthetic effect between siPNUTS treatment and lytic reactivation. This is expected for cells that are undergoing such robust KSHV lytic reactivation.

2.4. Line 171: the authors discuss expression of ORF50 after PNUTS knockdown, i.e. exclude a premature induction phenotype, but this experiment was performed under conditions of ectopic ORF50 expression. It is therefore not clear whether this interpretation would hold up in a more physiological context. It should also be explained how ectopic and viral ORF50 reads are distinguished in this experiment.

- We cannot distinguish these two transcripts in our analysis. This has been noted in the Extended Data Fig. 2 legend (lines 1269-1272) and in the revised text in the Results section (line 173). The RNA-seq traces show little ORF50 3' UTR (Extended data Fig 2c), so it seems most likely that the ectopically expressed ORF50 cDNA is the dominant transcript in these cells. Our primary concern was

that the high viral reactivation observed upon PNUTS knockdown could be an artifact of siPNUTS-induced overexpression of the ectopic ORF50; these data falsify that hypothesis. However, in TREx-BCBL1 cells lacking exogenously expressed ORF50, ORF50 increases upon PNUTS depletion (Fig 2i). Whether it is direct or indirect target remains unknown. Thus, we agree with the reviewer that ORF50 could be a PNUTS target in physiological contexts.

2.5. Details of TALEN-facilitated gene insertion should be cited or described (lines 524/525).

- Sanjana et al. 2012 has now been cited here (line 578).

Reviewer #3 (Remarks to the Author):

In their manuscript entitled "The PNUTS-protein phosphatase 1 complex acts as an intrinsic barrier to KSHV gene expression and replication", the authors identify the PNUTS-PP1 complex as a critical suppressor of lytic KSHV gene expression using a genome-wide CRISPR screen. They show that mechanistically PNUTS requires PP1 interaction. Surprisingly, PNUTS was also found to represses productive elongation at the 5'-end of the KSHV reporter gene.

The manuscript is well written and easy to follow. Besides the identification of PNUTS, the CRISPR screen provides exciting data on the involvement of other cellular genes with known roles in transcription. While the exact molecular mechanism remains unclear the discussion provides an excellent overview on the putative mechanism. The paper will be of importance for many groups in the field of transcription.

3.1. I have no major comments. The only minor aspect I found a bit irritating when I first read the paper was "PNUTS-protein phosphatase 1 (PP1) complex" in the abstract as PNUTS was abbreviated but PP1 was not. I would suggest to replace this by "PNUTS-PP1 complex" and only explain this in the main manuscript.

- This change has been made in the abstract and title (lines 1 and 24).

Reviewer comments, second round review –

Reviewer #1 (Remarks to the Author):

The authors have diligently addressed the issues raised in my review and found some additional interesting corroborative evidence of the effect of PNUTS on an endogenous KSHV gene. I have no further criticisms.

Reviewer #2 (Remarks to the Author):

This manuscript was carefully revised and my only comment would be to request an earlier description of the function (as in the discussion) and genome coordinates for the T1.4 transcript/Fig. 7A, since the DR1/DR2 terminology is also used for the K12 locus, which was initially confusing to me.

REVIEWERS' COMMENTS

Reviewer #1 (Remarks to the Author):

The authors have diligently addressed the issues raised in my review and found some additional interesting corroborative evidence of the effect of PNUTS on an endogenous KSHV gene. I have no further criticisms.

We thank the Reviewer and are happy to see our revisions have strengthened the manuscript's evidence.

Reviewer #2 (Remarks to the Author):

This manuscript was carefully revised and my only comment would be to request an earlier description of the function (as in the discussion) and genome coordinates for the T1.4 transcript/ Fig. 7A, since the DR1/DR2 terminology is also used for the K12 locus, which was initially confusing to me.

We have added an earlier description of the T1.4 transcript at its initial introduction in the Results section (p 17, bottom).